# Visually Consistent Hierarchical Image Classification

**Seulki Park**[1], **Youren Zhang**[1], **Stella X. Yu**[1,2], **Sara Beery**[3], **Jonathan Huang**[4]
[1]University of Michigan    [2]UC Berkeley    [3]MIT    [4]Scaled Foundations
{seulki,yourenz,stellayu}@umich.edu, beery@mit.edu, jhuang11@gmail.com

## Abstract

Hierarchical classification predicts labels across multiple levels of a taxonomy, e.g., from coarse-level *Bird* to mid-level *Hummingbird* to fine-level *Green hermit*, allowing flexible recognition under varying visual conditions. It is commonly framed as multiple single-level tasks, but each level may rely on different visual cues: Distinguishing *Bird* from *Plant* relies on *global features* like *feathers* or *leaves*, while separating *Anna's hummingbird* from *Green hermit* requires *local details* such as *head coloration*. Prior methods improve accuracy using external semantic supervision, but such statistical learning criteria fail to ensure consistent visual grounding at test time, resulting in incorrect hierarchical classification. We propose, for the first time, to enforce *internal visual consistency* by aligning fine-to-coarse predictions through intra-image segmentation. Our method outperforms zero-shot CLIP and state-of-the-art baselines on hierarchical classification benchmarks, achieving both higher accuracy and more consistent predictions. It also improves internal image segmentation without requiring pixel-level annotations.

## 1 Introduction

Hierarchical classification (Silla & Freitas, 2011; Chang et al., 2021; Jiang et al., 2024) predicts labels along a semantic taxonomy (e.g., *Bird → Hummingbird → Green hermit*), rather than choosing from a single flat label set (Fig. 1). This flexibility aligns well with real-world needs: a casual observer may only recognize a general category like *Bird*, while a biologist might identify the exact species. More importantly, when visual details are lacking such as *a bird seen from afar*, hierarchical models can still offer informative predictions (*Bird*), whereas flat fine-grained classifiers may fail entirely.

Hierarchical classification is routinely framed as multiple single-level tasks, but each level may depend on different visual cues: Distinguishing *Bird* from *Plant* relies on *global features* like feathers or leaves, while separating *Anna's hummingbird* from *Green hermit* requires *local details* such as head coloration. Our work is the first to enforce visual consistency within an image to improve both the accuracy of hierarchical classification and the consistency of predictions across levels (Fig. 1).

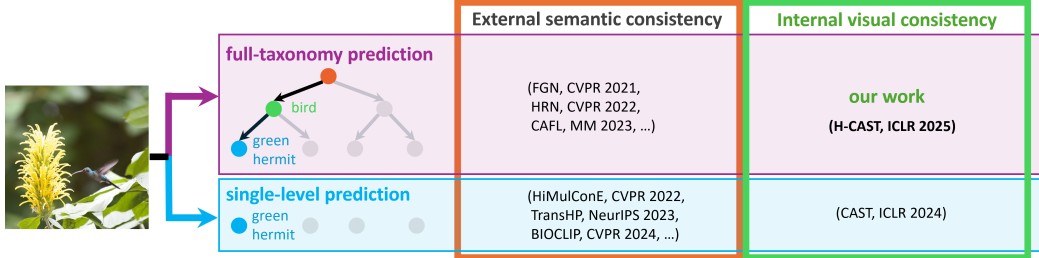

Figure 1: **We propose enforcing internal visual consistency to improve hierarchical classification across taxonomy levels.** Prior works rely on external semantic supervision, a statistical criterion that fails to ensure consistent visual focus at test time. Our approach is the first to align predictions through intra-image consistency, improving both accuracy and coherence. Our code is available at https://github.com/pseulki/hcast.

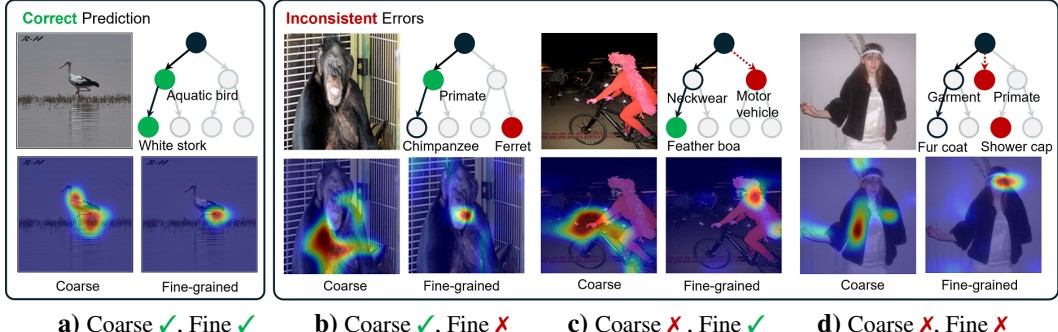

**a)** Coarse ✓, Fine ✓    **b)** Coarse ✓, Fine ✗    **c)** Coarse ✗, Fine ✓    **d)** Coarse ✗, Fine ✗

Figure 2: **Incorrect hierarchical classification often results from inconsistent visual grounding across hierarchy levels.** We show Grad-CAM visualizations (Selvaraju et al., 2017) of FGN (Chang et al., 2021) trained on BREEDS (Entity-30) (Santurkar et al., 2021). **a)** A consistent case: both classifiers focus on the same object, with the fine-grained classifier capturing details (*bird leg*), and the coarse classifier attending to the *whole bird*. **b)** The coarse classifier localizes the *chimpanzee* but the fine classifier fails to attend to its crucial details and makes a wrong prediction. **c)** The fine-grained classifier correctly identifies the *feather boa*, while the coarse classifier wrongly attends to the *bicycle*. **d)** Both classifiers attend to misaligned areas and make wrong predictions. These cases show that semantic accuracy relies on consistent visual grounding. Our model aligns visual attention across levels, capture different details within a coherent region, and predict all four cases correctly.

Existing works improve fine-grained accuracy by training with external, consistent semantic supervision across hierarchy levels (Chang et al., 2021; Chen et al., 2022; Wang et al., 2023a). However, *statistical alignment across levels during training* does not guarantee *visual consistency at test time*. Such inconsistencies worsen when multiple semantic concepts appear in an image, leading to conflicting predictions that violate the hierarchical taxonomy.

Using Grad-CAM visualizations (Selvaraju et al., 2017) on FGN (Chang et al., 2021) trained on the two-level BREEDS (Entity-30) dataset (Santurkar et al., 2021), we observe that incorrect predictions result from coarse and fine-grained classifiers attending to entirely different regions (Fig. 2). In Fig. 2c, the coarse classifier focuses on the *bicycle* and predicts *Motor Vehicle*, while the fine-grained classifier attends to the *headwear* and predicts *Feather Boa* instead. In contrast, when both levels focus on related regions, e.g., the *whole bird* and its *leg* in Fig. 2a, the predictions are consistent and correctly aligned with the taxonomy. See a quantitative validation of this observation in Appendix A.

We propose enforcing internal visual parsing consistency to achieve semantic coherence and thus more accurate hierarchical classification (Fig. 3). To recognize a *Green hermit*, the fine-grained classifier attends to the *beak*, *wings*, and *tail*, while the coarse classifier attends to their collective

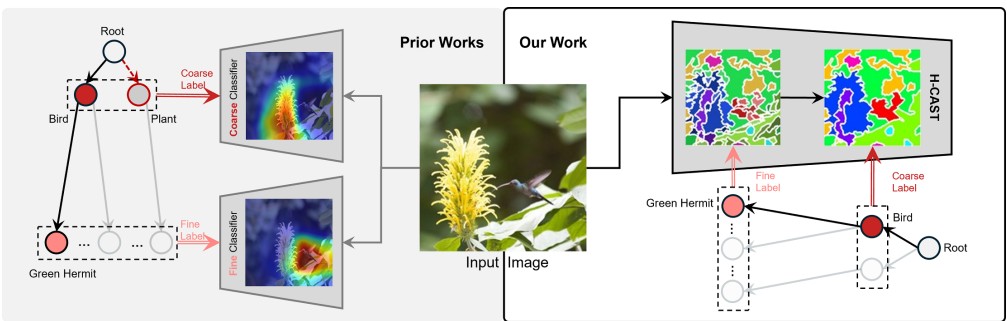

Figure 3: **Our method ensures internal visual consistency by aligning coarse and fine classifiers on hierarchical segmentation, unlike prior approaches that rely only on external semantic losses without visual grounding.** Segmentation outputs show how fine details (e.g., wings, head, tail) at the 32-way level are grouped into a unified bird region at the 8-way level. Identical color hues indicate consistent groupings, encouraging the model to attend to coherent image regions.

region to recognize the *Bird*. Our model progressively groups fine details into larger areas and shares features across levels, addressing inconsistencies in prior methods that treat each level independently.

We develop H-CAST, a supervised hierarchical extension of CAST (Ke et al., 2024), which leverages CAST's coarse-to-fine segmentation for hierarchical classification (Fig. 1). We apply fine- and coarse-grained classification losses at early and late stages, respectively, using Tree-path KL Divergence loss to jointly optimize accuracy across all levels. H-CAST ensures visual consistency with its architecture and propagates semantic errors across levels to produce aligned and accurate hierarchical predictions.

H-CAST outperforms state-of-the-art methods with 10% absolute improvement in Full-Path Accuracy (FPA) compared to ViT-based baselines on BREEDS, demonstrating superior consistency and robustness. FPA, our proposed metric, measures the proportion of samples correctly classified at all hierarchy levels. Even vision foundation models like CLIP (Radford et al., 2021) struggle with maintaining hierarchical consistency. Our extensive experiments demonstrate that, H-CAST not only significantly improves hierarchical classification, but also enhances internal image segmentation without requiring pixel-level annotations.

**Our contributions. 1)** We introduce **internal visual consistency**, aligning fine and coarse classification with consistent hierarchical segmentation to ensure classifiers across levels focus on coherent and related regions. **2)** We propose a **Tree-Path KL Divergence Loss** to enforce semantic consistency across hierarchy levels. **3)** Our method significantly outperforms ViT-based baselines, achieving +3–10% FPA across all datasets, demonstrating superior consistency and robustness.

## 2 RELATED WORK

**Hierarchical classification** problem presents a unique challenge: the image remains the same, but the output changes in the *semantic* (text) space (e.g. "Birds"→ "Hummingbird"→ "Anna's hummingbird"). Due to this structure, prior work has primarily focused on embedding data into the semantic space, using additional loss functions (Bertinetto et al., 2020; Zeng et al., 2022) or encoding entire taxonomies as lengthy text inputs, as in BIOCLIP (Stevens et al., 2024). *In contrast*, we approach the problem from the *visual space*, exploring how hierarchical classification relates to visual grounding by ensuring consistency across different levels of detail, from fine-grained features to broader features. **This visual-grounding perspective is novel and has not been explored in prior work** (see Fig. 1). Existing hierarchical classification methods can be divided into two approaches:

**1. Single-level prediction:** Single-level prediction can be further divided into two approaches. First, fine-grained classification (bottom-up) targets fine-grained classes (leaf nodes), based on the assumption that coarse categories can be inferred using a predefined taxonomy (Karthik et al., 2021; Zhang et al., 2022; Wang et al., 2023b; Stevens et al., 2024). While effective for clear images, it struggles with ambiguous test-time conditions when fine-grained classification is impossible (e.g., birds at high altitudes). Second, the local classifier (top-down) predicts a single label at the most confident level by refining coarse-to-fine predictions (Deng et al., 2010; Wu et al., 2020; Brust & Denzler, 2019). However, errors from coarse levels can propagate downward, limiting accuracy. We address this by predicting across the entire taxonomy for greater robustness.

**2. Full-taxonomy prediction:** predicts all levels simultaneously using a shared backbone with separate branches (Zhu & Bain, 2017; Wehrmann et al., 2018; Chang et al., 2021; Liu et al., 2022). A key challenge is maintaining label consistency. While Wang et al. (2023a) adjusted predictions to enforce consistency, separate branches process images independently, leading to misalignment. To address this, we propose a model based on consistent visual grounding. To the best of our knowledge, no prior work has utilized visual segments to resolve inconsistency in hierarchical classification.

**Unsupevised/Weakly-supervised Semantic Segmentation** aims to group pixels without pixel-level annotations or using only class labels (Hwang et al., 2019; Ouali et al., 2020; Ke et al., 2022; 2024). These works employ hierarchical grouping to achieve meaningful segmentation *without* pixel-level labels. In this field, "hierarchical" refers to part-to-whole visual grouping, where smaller units (e.g., a person's face or arm) are grouped into larger regions (e.g., the whole body). Based on our intuition that fine-grained classifiers need more detailed information, while coarse classifiers focus on broader groupings, our approach leverages these varying types of visual grouping. To implement this, we adopt the recently proposed CAST (Ke et al., 2024), whose graph pooling naturally supports consistent visual grouping. Notably, our work introduces the novel insight that **part-to-whole spatial**

**granularity can align with taxonomy hierarchies** (e.g., finer segments for fine-grained labels, coarser segments for coarse labels), a connection not previously explored.

Detailed related work, including **Hierarchical Semantic Segmentation** is included in Appendix B.

## 3 CONSISTENT HIERARCHICAL CLASSIFICATION

We improve the consistency and accuracy of hierarchical classification via a progressive learning scheme, where each level builds upon the previous one instead of training separate models per level (Fig. 4). We address two key inconsistencies: *visual inconsistency*, where classifiers attend to different regions (Fig.3), and *semantic inconsistency*, where predictions violate the taxonomy (e.g., Plant"–Hummingbird"). We propose H-CAST (Sec. 3.1) for visual consistency, and a novel Tree-path KL Divergence loss that encodes parent-child relations to enforce semantic alignment (Sec. 3.2).

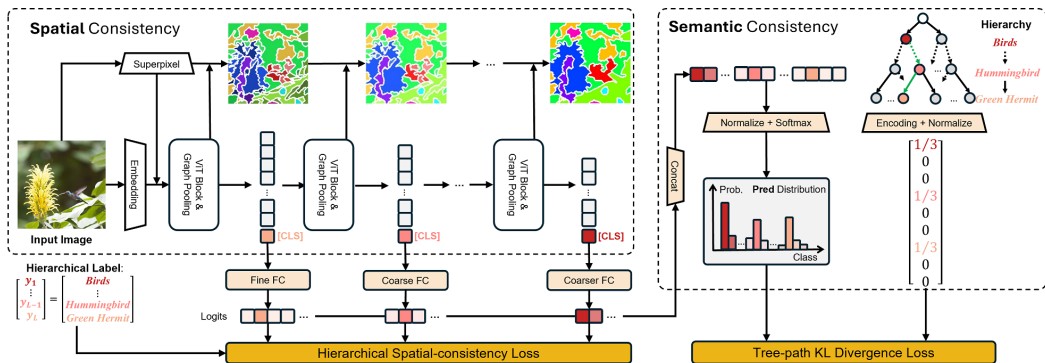

Figure 4: **Our method implements visually grounded hierarchical classification through visual and semantic modules.** The **Visual Consistency** module uses fine-to-coarse superpixel groupings to ensure classifiers at different levels focus on corresponding regions while capturing different details. The **Semantic Consistency** module encodes label hierarchies to align predictions across levels. Together, they encourage cooperative learning across the hierarchy and improve overall performance.

### 3.1 H-CAST FOR VISUAL CONSISTENCY

The areas of focus within the image need to differ when conducting classification at the fine-grained level compared to the coarse level. When distinguishing between similar-looking species (e.g., "Green Hermit" vs. "Anna's Hummingbird"), the fine-grained recognition requires attention to *fine details* like the bird's beak and wings; meanwhile, at the coarse level (e.g., "bird" vs. "plant"), the attention shifts to *larger parts* such as the overall body of the bird. However, this shift in focus towards larger objects does not imply a sudden disregard for the previously focused details and a search for new larger objects. Rather, a natural approach involves combining detailed features such as the bird's beak, belly, and wings for accurate bird recognition. Therefore, we argue that **the hierarchical model should be grounded in consistent visual cues**. From this insight, we design a model where the details learned at the fine level (e.g., bird's beak and wings) are transferred to the coarse level as broader parts (e.g., bird's body) through consistent feature grouping.

For internally consistent feature grouping, we build upon recent work CAST (Ke et al., 2024). CAST develops a hierarchical segmentation from fine to coarse, an internal part of the recognition process. However, their segmentation is driven by a flat recognition objective at the very end of visual parsing. We extend it by imposing fine-to-coarse semantic classification losses at different stages of segmentations throughout the visual parsing process. Our design reflects the intuition that finer segments can be helpful in capturing fine-grained details (e.g., beaks and wings) required for fine-grained recognition, whereas coarser segments can be effective in representing broader features (e.g., the body of a bird) needed for coarse-grained recognition. We have a single hierarchical recognition grounded on internally consistent segmentations, each driven by a classification objective at a certain granuality. We refer to our method as *Hierarchical-CAST* (H-CAST).

Consider a hierarchical recognition task where $x$ denotes an image associated with hierarchical labels $y_1, \ldots y_L$, encompassing a total of $L$ levels in the hierarchy. Level $L$ is the finest level (*i.e.,* leaf node), and Level 1 is the coarsest level (*i.e.,* root node). Then, given an image $x$, the hierarchical image recognition task is to predict labels at all levels across the hierarchy.

Let $Z_l$ and $S_l$ denote the feature and segment representations at the $l$-th level. We extract superpixels from image $x$ using SEEDS (Van den Bergh et al., 2012), which groups pixels by color and local connectivity. These superpixels serve as ViT input in place of fixed patches and form the initial (finest) segments $S_{L+1}$. Each feature $Z_l$ consists of class tokens ($Z_l^{class}$) and segment tokens ($Z_l^{seg}$). Graph pooling (Ke et al., 2024) then merges similar segments, enabling features to capture increasingly global context from $Z_L$ to $Z_1$. For hierarchical recognition, we add a classification head ($f_l$) consisting of a single linear layer at each level. Then, we define the hierarchical visual consistency loss as the sum of $L$ cross-entropy losses ($L_{CE}$), denoted as

$$L_{HV} = \sum_{l=1}^{L} L_{CE}(f_l(Z_l^{class}), y_l). \tag{1}$$

H-CAST applies hierarchical supervision across all levels, unlike CAST, which uses flat supervision only on the final class token. This design allows labels at different levels to inform one another. In Sec. 4.6, we show its effectiveness over alternatives, including reverse coarse-to-fine supervision.

## 3.2 TREE-PATH KL DIVERGENCE LOSS FOR SEMANTIC CONSISTENCY

To improve semantic consistency across hierarchy levels, we propose Tree-path KL Divergence loss that imposes tree path relationships between labels. We first concatenate labels from all levels to create a distribution, as $Y = \frac{1}{L}[1_{y_L}; \ldots; 1_{y_1}]$, where $1_{y_l}$ represents the one-hot encoding for level $l$. Next, we concatenate the outputs of each classification head and then apply the log softmax function (LogSoftmax). We use Kullback–Leibler divergence loss ($KL$) to align this output with the ground truth distribution $Y$. Then, TK loss is calculated as follows.

$$L_{TK} = KL(\text{LogSoftmax}([f_L(Z_L^{class}); \ldots; f_1(Z_1^{class})]), Y). \tag{2}$$

This loss penalizes predictions that do not align with the taxonomy by simultaneously training on multiple labels within the hierarchy. Therefore, despite the simplicity, TK loss enables the model to enhance semantic consistency through this vertical encoding from the root (parent) node of the hierarchy level to the leaf (children) node. Our final loss becomes as follows, where $\alpha$ is a hyperparameter to control the weight of $L_{TK}$,

$$L = L_{HV} + \alpha L_{TK}. \tag{3}$$

## 4 EXPERIMENTS

We first show that hierarchical classification remains challenging—even for vision foundation models, which often yield inconsistent predictions. Our method outperforms existing approaches and flat baselines on benchmark datasets. We further validate our design through ablations and demonstrate that hierarchical supervision also benefits semantic segmentation.

## 4.1 EXPERIMENTAL SETTINGS

**Datasets.** We use widely used benchmarks in hierarchical recognition: BREEDS (Santurkar et al., 2021), CUB-200-2011 (Welinder et al., 2010), FGVC-Aircraft (Maji et al., 2013), and iNat21-Mini (Van Horn et al., 2021). BREEDS, a subset of ImageNet (Russakovsky et al., 2015), includes four 2-level hierarchy datasets with different depths/parts based on the WordNet (Miller, 1995) hierarchy: Living-17, Non-Living-26, Entity-13, Entity-30. For BREEDS, we conduct training and validation using their source splits. BREEDS provide a wider class variety and larger sample size than CUB-200-2011 and FGVC-Aircraft, making it better suited for evaluating generalization performance. CUB-200-2011 comprises a 3-level hierarchy with order, family, and species; FGVC-Aircraft consists of a 3-level hierarchy including maker, family, and model (*e.g.*, Boeing - Boeing 707 - 707-320 ); For experiments on a larger dataset, we used the 3-level iNat21-Mini. Details of the iNat21-Mini are provided in Sec. E.4. Table 4 in Appendix summarizes a description of the datasets.

**Evaluation Metrics.** We evaluate our models using metrics for both accuracy and consistency.

- **level-accuracy**: the proportion of correctly classified instances at each level (Chang et al., 2021).
- **weighted average precision (wAP)** (Liu et al., 2022): wAP $= \sum_{l=1}^{L} \frac{N_l}{\sum_{k=1}^{L} N_k} P_l$, where $N_l$ and $P_l$ denote the number of classes and Top-1 classification accuracy at level $l$, respectively. This metric considers the classification difficulty across different hierarchies.
- **Tree-based InConsistency Error rate (TICE)** (Wang et al., 2023a): TICE $= n_{ic}/N$, where $n_{ic}$ denotes the number of samples with inconsistent prediction paths, and $N$ refers to the number of all test samples. This tests whether the prediction path exists in the tree (consistency).
- **Full-Path Accuracy (FPA)**: FPA $= n_{ac}/N$, where $n_{ac}$ refers to the number of samples with all level of labels correctly predicted. This metric evaluates **both accuracy and consistency**, **ultimately representing our primary metric of interest**.

The difference between FPA and TICE is illustrated in Table 6 in Appendix.

**Comparison methods.** First, we evaluate our H-CAST with representative models in hierarchical classification, **FGN** (Chang et al., 2021) and **HRN** (Chen et al., 2022). **FGN** uses level-specific heads to avoid negative transfer across granularity levels, while **HRN** employs residual connections to capture label relationships and a hierarchy-based probabilistic loss. We also compare **TransHP** (Wang et al., 2023b), a ViT-based model that learns prompt tokens to represent coarse classes and injects them into an intermediate block to enhance fine-grained predictions. Lastly, we compare **Hier-ViT**, a variant without visual segments and TK loss. Like our approach, **Hier-ViT** trains each hierarchy level using the class token from the last $6, 9, 12$ blocks. To establish a *ceiling baseline*, we compare with flat models trained at a single hierarchy level. **Flat-ViT** classifies one level using the ViT class token, while **Flat-CAST** trains independent models for each level using the CAST architecture (Ke et al., 2024). We also compare with **Hierarchical Ensembles, HiE** (Jain et al., 2023), which improves fine-grained predictions via post-hoc correction using a coarse model. Note that flat models require separate models for each hierarchy level, leading to increased storage and training costs. For baseline methods, we use the official codebases and their reported optimal settings. All models are trained for 100 epochs, except TransHP, which is trained for 300 epochs as in the original paper. Details on the architecture and hyperparameter settings for H-CAST can be found in Appendix D.

## 4.2 HIERARCHICAL CLASSIFICATION WITH VISION FOUNDATION MODELS

First, to demonstrate that longstanding hierarchical classification is not easily solved by today's vision foundation models, we evaluate CLIP (Radford et al., 2021)'s performance on the 2-level BREEDS dataset. The top row of Fig. 5 shows prediction rates on the test set, while the bottom row presents

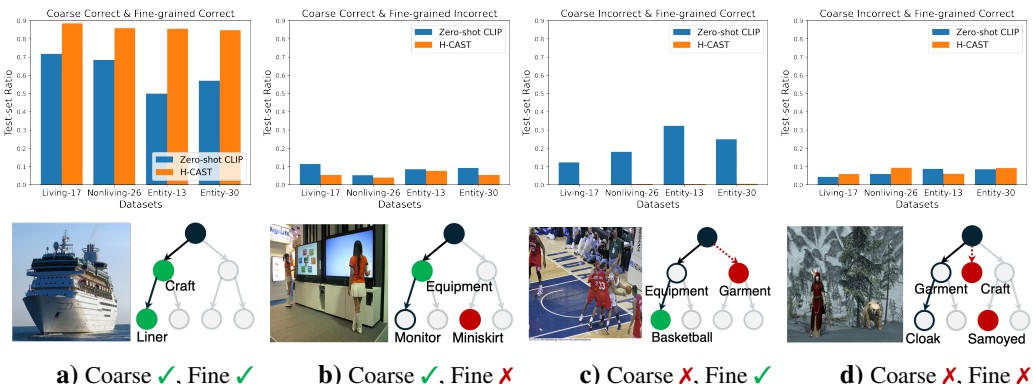

**a)** Coarse ✓, Fine ✓    **b)** Coarse ✓, Fine ✗    **c)** Coarse ✗, Fine ✓    **d)** Coarse ✗, Fine ✗

Figure 5: **Vision foundation models struggle with consistent predictions in hierarchical classification.** We evaluate CLIP (Radford et al., 2021) on the 2-level BREEDS dataset (top) and present misclassification examples from Entity-13 (bottom). **a)** CLIP struggles to maintain consistency and correctness, achieving only about 50% accuracy on Entity-13. **b)** CLIP more frequently predicts the coarse category correctly while misclassifying the fine-grained category compared to H-CAST across all datasets. **c)** CLIP often predicts the fine-grained category correctly but fails at the coarse level, a mistake that is rare in H-CAST, suggesting difficulty in grasping broader conceptual understanding. H-CAST accurately predicts cases **a-c**. **d)** Both CLIP and H-CAST fail in complex scenes.

examples from the Entity-13 dataset. Even considering the zero-shot prediction, Fig. 5 (a) shows that the overall ratio of correct predictions for both coarse and fine-grained classifications is significantly low, with only around 50% accuracy on the Entity-13 dataset. Fig. 5 (c) further highlights significant errors in coarse predictions when addressing broader concepts. Our findings support the recent study from Xu et al. (2024) that VLMs excel at fine-grained prediction but struggle with general concepts. This highlights the ongoing challenges of hierarchical classification, even with vision foundation models. Furthermore, when we examine the misclassification examples in bottom row of Fig. 5, we can see that CLIP focuses on different object areas for coarse and fine-grained predictions. For example, in (b), CLIP predicts "Equipment" in the coarse prediction but predicts "Miniskirt" in the fine-grained prediction instead of "Monitor" (a child of Equipment). However, our model, based on consistent visual segments, can make correct predictions in all cases.

## 4.3 Consistent Hierarchical Classification on Benchmarks

Table 1 shows benchmarks on BREEDS. H-CAST outperforms both hierarchical (FGN, HRN, TransHP, Hier-ViT) and flat (ViT, CAST, HiE) baselines. It exceeds ViT-based models like Hier-ViT and TransHP by over 11 points, highlighting the effectiveness of our visual grounding and Tree-Path Loss beyond simply adding hierarchy supervision. While TransHP uses coarse labels as prompts to aid fine-grained prediction, it does not jointly optimize both levels, resulting in lower consistency. H-CAST also improves FPA by 4.3–6.4 points over HRN, despite having far fewer parameters.

Table 1: **H-CAST achieves both high consistency and accuracy, outperforming both hierarchical and flat baselines on BREEDS.** It achieves a 4.3-6.4 percentage point gain in FPA metric over HRN with significantly fewer parameters. Additionally, H-CAST surpasses Hier-ViT and TransHP by over 11 percentage points, demonstrating that its success is due to our consistent visual grounding and Tree-path Loss, rather than adding hierarchy supervision to a ViT backbone. (Higher the metric is the best, except TICE.) 'ViT-S' refers to ViT-Small, while 'RN-50' denotes ResNet-50.

| | | Configuration | | Living-17 (17-34) | | | | | Non-Living-26 (26-52) | | | | |
| | | backbone | #params | FPA | coarse | fine | wAP | TICE | FPA | coarse | fine | wAP | TICE |
|---|---|---|---|---|---|---|---|---|---|---|---|---|---|
| Flat | Flat-ViT | ViT-S | 65.0M | 66.24 | 75.71 | 72.06 | 73.28 | 17.11 | 57.46 | 67.50 | 65.73 | 57.46 | 23.27 |
| | Flat-ViT + HiE | ViT-S | 65.0M | 67.59 | 75.71 | 71.35 | 72.81 | 9.88 | 59.73 | 67.50 | 65.31 | 66.04 | 13.69 |
| | Flat-CAST | ViT-S | 78.5M | 78.82 | 88.06 | 82.88 | 84.61 | 8.82 | 76.17 | 84.77 | 81.08 | 82.31 | 11.77 |
| | Flat-CAST + HiE | ViT-S | 78.5M | 81.59 | 88.06 | 83.24 | 84.85 | 5.18 | 79.23 | 84.77 | 81.39 | 82.51 | 6.19 |
| Hierarchy | FGN | RN-50 | 24.8M | 63.82 | 72.59 | 68.00 | 69.53 | 12.12 | 60.81 | 69.46 | 65.77 | 67.00 | 16.46 |
| | HRN | RN-50 | 70.8M | _79.18_ | _87.53_ | _81.47_ | _83.49_ | _6.29_ | _76.31_ | _82.38_ | _80.15_ | _80.90_ | _9.54_ |
| | Hier-ViT | ViT-S | 21.7M | 74.06 | 80.94 | 74.88 | 76.90 | 10.50 | 72.04 | 73.31 | 68.39 | 70.03 | 12.45 |
| | TransHP | ViT-S | 21.7M | 74.35 | 83.00 | 76.65 | 78.76 | 8.35 | 68.62 | 77.31 | 72.31 | 73.97 | 13.04 |
| | Ours (H-CAST) | ViT-S | 26.2M | **85.12** | **90.82** | **85.24** | **87.10** | **3.19** | **82.67** | **87.89** | **83.15** | **84.73** | **5.26** |
| | *Our Gains over SOTA* | | | +5.94 | +3.29 | +3.77 | +3.61 | +3.10 | +6.36 | +5.51 | +3.00 | +3.83 | +4.28 |

| | | Configuration | | Entity-13 (13-130) | | | | | Entity-30 (30-120) | | | | |
| | | backbone | #params | FPA | coarse | fine | wAP | TICE | FPA | coarse | fine | wAP | TICE |
|---|---|---|---|---|---|---|---|---|---|---|---|---|---|
| Flat | Flat-ViT | ViT-S | 65.0M | 64.22 | 76.28 | 76.06 | 76.08 | 21.33 | 66.93 | 76.28 | 74.35 | 74.77 | 18.75 |
| | Flat-ViT + HiE | ViT-S | 65.0M | 65.20 | 76.47 | 74.91 | 75.05 | 15.68 | 68.77 | 76.47 | 73.92 | 74.43 | 11.08 |
| | Flat-CAST | ViT-S | 78.5M | 78.63 | 87.80 | 83.72 | 84.09 | 10.65 | 82.67 | 87.89 | 83.15 | 84.73 | 5.26 |
| | Flat-CAST + HiE | ViT-S | 78.5M | 79.52 | 87.80 | 83.40 | 83.80 | 6.83 | 83.70 | 87.89 | 84.30 | 85.02 | 4.20 |
| Hierarchy | FGN | RN-50 | 24.8M | 74.23 | 85.35 | 78.00 | 78.67 | 9.43 | 68.52 | 77.47 | 73.18 | 74.04 | 13.62 |
| | HRN | RN-50 | 70.8M | _81.43_ | _90.00_ | _84.48_ | _84.98_ | 6.34 | _79.85_ | _86.57_ | _83.35_ | _83.99_ | _8.38_ |
| | Hier-ViT | ViT-S | 21.7M | 74.63 | 86.95 | 75.39 | 77.70 | _5.19_ | 73.01 | 81.38 | 74.10 | 74.76 | 11.61 |
| | TransHP | ViT-S | 21.7M | 73.45 | 86.28 | 76.23 | 77.14 | 8.80 | 72.00 | 80.78 | 75.63 | 76.66 | 12.20 |
| | Ours (H-CAST) | ViT-S | 26.2M | **85.68** | **93.42** | **86.15** | **87.60** | **1.69** | **84.83** | **90.23** | **85.45** | **85.88** | **2.57** |
| | *Our Gains over SOTA* | | | +4.25 | +3.42 | +1.67 | +2.62 | +4.65 | +4.98 | +3.66 | +2.10 | +1.89 | +5.81 |

Note that flat models train separate networks per level, incurring higher memory and training costs. H-CAST outperforms them, demonstrating its efficiency and effectiveness for hierarchical classification. Similar results are observed on Aircraft and CUB (Appendix E.5), with additional evaluation on the larger iNaturalist 2021-Mini dataset in Appendix E.4.

## 4.4 Visualizations of Structured Visual Parsing

**Well-structured visual parsing is crucial for accurate hierarchical classification.** H-CAST improves hierarchical classification by ensuring structured visual parsing, where fine details progressively merge into meaningful objects at coarser levels. Fig. 6 visualizes this process on the

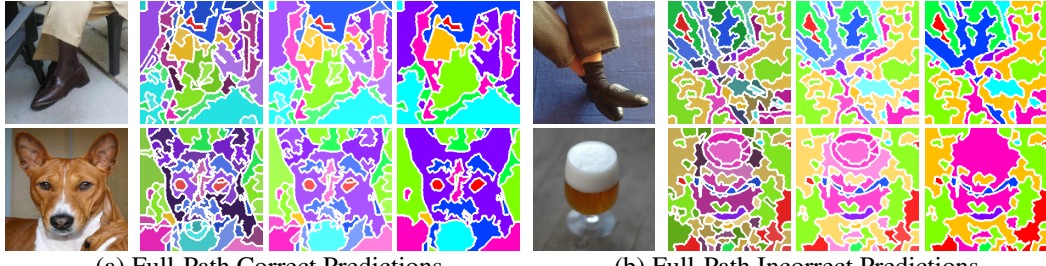

(a) Full-Path Correct Predictions          (b) Full-Path Incorrect Predictions

Figure 6: **Well-structured visual parsing enables accurate hierarchical classification, while fragmented segmentation correlates with misclassification.** We compare fine-to-coarse segmentation for fully correct (a) and incorrect (b) predictions on the Entity-30 dataset to show how visual parsing quality affects classification. For correct predictions (**a**), fine details merge into coherent objects at coarser levels, preserving structure. For example, in the correctly predicted *shoe* image, the *shoe* and *ankle* form a unified segment, maintaining object integrity. In contrast, for incorrect predictions (**b**), segmentation is fragmented and inconsistent. The misclassified *shoe* image shows scattered segments blending with the background, making it difficult to recognize the overall shape. Likewise, the *beer glass* lose key features due to unstable grouping. These results highlight that accurate hierarchical classification relies on structured visual parsing.

BREEDS Entity-30 dataset. In full-path correct predictions (Fig. 6, Left), fine-level details are effectively grouped into coherent objects. For example, in a correctly predicted *shoe* image, the shoe and ankle merge into a single segment, maintaining object integrity. Similarly, in the dog example, fine details in the earlier segments merge into a unified face, eyes, and nose at the coarse level, demonstrating structured feature grouping. Conversely, in full-path incorrect predictions (Fig. 6, Right), segmentation is fragmented and inconsistent. The misclassified shoe image shows scattered segments blending with the background, while even a simple-shaped object like a beer glass is broken into disconnected parts, losing its structural coherence. This highlights that structured visual parsing is key to accurate classification, while fragmented segmentation leads to errors.

## 4.5 Effect of Visual Grounding on Hierarchical Classification

**H-CAST Ensures Consistent and Class-Relevant Feature Learning.** We compare nearest neighbors of class token features at fine and coarse levels for queries correctly classified at both levels on Entity-30 (Fig. 7). If a model truly learns discriminative class features, its nearest neighbors should remain consistent across levels. However, Hier-ViT often retrieves visually similar but semantically incorrect images, while H-CAST consistently selects class-relevant neighbors at both levels. For example, Hier-ViT retrieves a person in a black uniform as the closest fine-level neighbor for an *Italian Greyhound*, likely due to clothing color similarity (top left). Similarly, for a curly-haired *Bedlington Terrier*, it selects a fur coat (bottom left), suggesting it relies on superficial textures rather than class-specific features. In contrast, H-CAST consistently retrieves the correct class, selecting a visually similar *Italian Greyhound* and *Bedlington Terrier* at both levels, ensuring more stable and meaningful feature learning. These results show that Hier-ViT lacks visual consistency across levels, while H-CAST maintains class-relevant and hierarchically aligned retrievals.

## 4.6 Ablation Analysis of Architecture Design and Loss Function in H-CAST

**Fine-to-Coarse vs. Coarse-to-Fine learning.** Our model adopts a Fine-to-Coarse (F→C) learning strategy, first learning fine labels in the lower block and progressively integrating coarser labels. This contrasts with prior methods, which typically learn coarse features first (Zhu & Bain, 2017; Yan et al., 2015; Wang et al., 2023b). To evaluate its effectiveness, we compare F→C with two baselines: Coarse-to-Fine (C→F), which follows a conventional hierarchy by learning coarse labels first, and Fine-Coarse Merging (C+F), which combines fine and coarse features across blocks. For fairness, we exclude Tree-path KL Divergence loss in these comparisons. Table 2a (FGVC-Aircraft) shows that C→F yields the lowest fine-grained accuracy, while C+F slightly improves accuracy but adds significant parameter overhead. In contrast, F→C balances simplicity and strong performance,

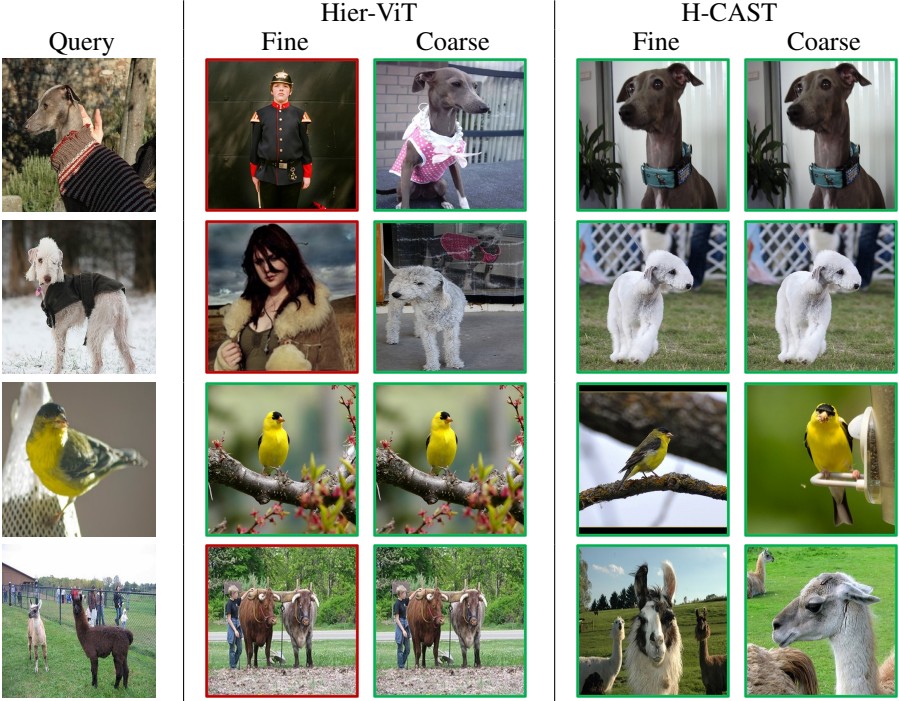

Figure 7: **Consistent visual grounding enhances feature learning in hierarchical classification.** We compare nearest neighbors of class token features at fine and coarse levels for queries correctly classified by H-CAST at both levels (Entity-30). **Rows 1,2)** Hier-ViT retrieves a person in a black uniform or black hair at the fine level. This raises doubts about whether the model truly learned the dog's features. **Row 3)** The query bird has gray wings with a yellow body. H-CAST retrieves a gray-winged bird at the fine level and a yellow-bodied bird at the coarse level, capturing fine-level detail and coarse-level semantics. Hier-ViT lacks this distinction. **Row 4)** The query shows two llamas on a green field. Hier-ViT retrieves two oxen in a similar setting, resulting in fine-level misclassification. H-CAST retrieves true llama images: one showing both white and brown llamas at the fine level, and a typical llama at the coarse level. This shows that H-CAST attends to class-relevant features while maintaining hierarchical consistency. Green image borders indicate same-class retrievals; red image borders indicate mismatches.

making it an effective choice for hierarchical classification. Additionally, attention visualizations (Appendix E.2) reveal that lower blocks focus on fine details, while upper blocks capture broader structures, demonstrating that our design effectively guides attention across hierarchy levels.

**Ablation Studies on the Proposed Losses.** We conduct two ablation studies to evaluate our loss functions. First, we assess the individual contributions of Hierarchical Spatial-consistency loss $L_{HS}$

Table 2: **Ablation studies of learning design and loss functions on Aircraft data.**

| Direction | C→F | C+F | F→C | Loss abl. | $L_{TK}$ | $L_{HS}$ | Both | Consis. | Flat. | BCE | KL Div. |
|-----------|-----|-----|-----|-----------|----------|----------|------|---------|-------|-----|---------|
| FPA | 82.01 | 81.76 | **82.66** | FPA | 82.48 | 82.66 | **83.72** | FPA | 82.87 | 82.18 | **83.72** |
| maker | 93.16 | 93.52 | **94.27** | maker | 94.30 | 94.27 | **94.96** | maker | 94.63 | 94.21 | **94.96** |
| family | 89.92 | **90.31** | 90.19 | family | 90.37 | 90.19 | **91.39** | family | 90.94 | 90.13 | **91.39** |
| model | 84.10 | **84.58** | 84.40 | model | 84.04 | 84.40 | **85.33** | model | 84.97 | 84.88 | **85.33** |
| wAP | 87.50 | **87.93** | 87.91 | wAP | 87.80 | 87.91 | **88.90** | wAP | 88.51 | 88.11 | **88.90** |

| (a) **'Coarse→Fine' design achieves best performance.** | (b) **Utilizing both losses yields best performance.** | (c) **KL Divergence loss outperforms alternative losses.** |
|---|---|---|

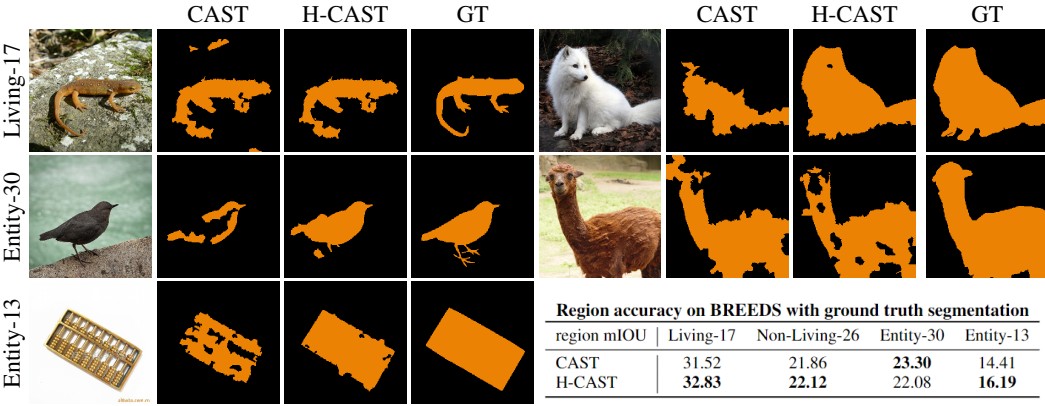

Figure 8: **H-CAST improves segmentation by leveraging hierarchical taxonomy.** We visualize segmentation results on BREEDS dataset and measure the region mIOU of fine-level objects for samples with segmentation ground truth (GT) from ImageNet-S (Gao et al., 2022). In the visualized images, we can observe that H-CAST better captures the overall shape in a more coherent manner compared to CAST. In quantitative evaluation, H-CAST outperforms CAST in most datasets despite using coarse-level supervision for the last-level segments whereas CAST employs fine-level supervision. It is surprising to find that the taxonomy hierarchy can help part-to-whole segmentation.

and Tree-path KL Divergence loss $L_{TK}$ on Aircraft. Table 2b shows that both losses significantly enhance performance, with their combination achieving the best accuracy and consistency. Next, we examine the effect of different loss functions by replacing KL Divergence loss with Binary Cross Entropy (BCE) and Flat Consistency loss. BCE directly substitutes the KL divergence component, while Flat Consistency loss, inspired by a bottom-up approach, infers coarse labels from fine predictions using BCE. As shown in Table 2c, KL Divergence loss achieves the highest FPA, demonstrating superior accuracy and consistency. Additional results on Living-17 are in Appendix E.3.

## 4.7 Additional Benefits of Hierarchical Classification for Segmentation

**Hierarchical semantic recognition enhances segmentation.** Although our primary focus is hierarchical recognition, we investigate whether incorporating hierarchical label information can also improve segmentation. Fig. 8 provide a qualitative and quantitative comparison between H-CAST and CAST. H-CAST, which uses varying granularity supervision for segments, outperforms CAST, which employs fine-grained level supervision, on most datasets such as Living-17, Non-Living-26, and Entity-13. The visualized results show that H-CAST better captures the overall shape in a more coherent manner compared to CAST. These findings demonstrate that utilizing hierarchical taxonomy benefits not only recognition but also segmentation. Details of the evaluation method and more visualization comparison with CAST are included in the Appendix E.7 and Fig. 11.

## 5 Summary

We tackle inconsistent predictions in hierarchical classification by introducing consistent visual grounding, leveraging varying granularity segments to align coarse and fine-grained classifiers. Unlike existing methods that soly rely on external semantic constraints, our approach leverages varying granularity segments to guide hierarchical classifiers, ensuring they focus on coherent and relevant regions across all levels. This leads to significant improvements across benchmarks, setting a new standard for robust hierarchical classification.

## Acknowledgements

This project was supported, in part, by NSF 2215542, NSF 2313151, a Berkeley AI Research grant with Google, and Bosch gift funds to S. Yu at UC Berkeley and the University of Michigan, with partial compute support from NAIRR Pilot (CIS240431, CIS250430).

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

# Appendix

## A  QUANTITATIVE EVIDENCE FOR CONSISTENT VISUAL GROUNDING

To quantitatively validate our observation that inconsistent predictions often occur when coarse and fine-grained classifiers focus on different regions in Figure 3, we analyze the Grad-CAM (Selvaraju et al., 2017) heatmaps of these classifiers. Specifically, we compute two metrics: the overlap score and the correlation score.

The **overlap score** quantifies the degree to which the regions activated by the two classifiers coincide. For each heatmap, we define a significant region as the set of pixels where activation values exceed a threshold. Specifically, the overlap count ($O$) measures the number of overlapping pixels between heatmaps $A$ and $B$, where both values exceed a threshold ($\tau = 0.001$). It is defined as:

$$O = \sum_{i,j} \left[ M_A(i,j) \wedge M_B(i,j) \right], \tag{4}$$

where $M_A(i,j)$ and $M_B(i,j)$ are binary masks indicating significant regions in $A$ and $B$, respectively. These masks are defined as:

$$M_A(i,j) = \begin{cases} 1 & \text{if } A(i,j) > \tau, \\ 0 & \text{otherwise,} \end{cases} \quad M_B(i,j) = \begin{cases} 1 & \text{if } B(i,j) > \tau, \\ 0 & \text{otherwise.} \end{cases} \tag{5}$$

The **correlation score** measures the linear relationship between the activation values of the overlapping regions in the two heatmaps. Let $A_k$ and $B_k$ the values in the overlapping regions, then correlation score is computed as:

$$R = \frac{\sum_{k=1}^{n}(A_k - \mu_A)(B_k - \mu_B)}{\sqrt{\sum_{k=1}^{n}(A_k - \mu_A)^2 \cdot \sum_{k=1}^{n}(B_k - \mu_B)^2}}, \tag{6}$$

where $n$ is the number of overlapping pixels, $\mu_A$ is the mean of $\{A_k\}$, and $\mu_B$ is the mean of $\{B_k\}$.

Higher overlap and correlation scores indicate stronger agreement between the regions attended to by the two classifiers. Conversely, lower scores highlight a lack of alignment in their focus.

Interestingly, empirical results from the FGN model (Chang et al., 2021) on the Entity-30 dataset show that when both classifiers make correct predictions, the overlap and correlation scores are significantly higher. In contrast, incorrect predictions correspond to notably lower scores, as shown in Table 3. These findings support our hypothesis that aligning the focus of coarse- and fine-grained classifiers enhances both prediction accuracy and consistency.

Table 3: **Overlap and correlation scores between coarse and fine-grained Grad-CAM heatmaps.** This shows that correct predictions correspond to higher overlap and correlation between coarse and fine-grained classifiers, highlighting the importance of aligning classifier focus for accuracy and consistency.

| Overlap | | Fine-grained | |
|---|---|---|---|
| | | True | False |
| Coarse | True | **0.51 ± 0.20** | 0.25 ± 0.13 |
| | False | 0.36 ± 0.18 | 0.37 ± 0.19 |

**(a)** Overlap Scores

| Correlation | | Fine-grained | |
|---|---|---|---|
| | | True | False |
| Coarse | True | **0.70 ± 0.26** | -0.02 ± 0.40 |
| | False | 0.30 ± 0.42 | 0.35 ± 0.41 |

**(b)** Correlation Scores

## B  ADDITIONAL RELATED WORK

**Hierarchical classification** can be divided into **single-level prediction** and **full-taxonomy prediction** based on the output structure. Single-level prediction is further categorized into **fine-grained classification** and **local classifier** approaches.

**1) Single-level prediction:**
**1-1) The fine-grained classification (bottom-up) approach** focuses on predicting fine-grained classes (e.g., leaf nodes) by leveraging taxonomy (Deng et al., 2014; Zhang et al., 2022; Zeng et al., 2022). It is often referred to as a bottom-up method because higher-level coarse classes can be inferred from the predicted fine-grained classes. Various methods have been proposed to effectively use hierarchical information. For example, hierarchical cross-entropy (HXE) loss (Bertinetto et al., 2020) reweights cross-entropy terms along the hierarchy tree based on class depth. Inspired by transformer prompting techniques, TransHP (Wang et al., 2023b) introduced coarse-class prompt tokens to improve fine-grained classification accuracy. Recently, BIOCLIP (Stevens et al., 2024), trained on large-scale Tree of Life data, achieved superior few-shot and zero-shot performance using a CLIP (Radford et al., 2021) contrastive objective on text combining fine-grained and higher-level classes. One of the actively studied topics is minimizing "mistake severity" (e.g., the tree distance between incorrect predictions and the ground truth) (Bertinetto et al., 2020; Karthik et al., 2021; Garg et al., 2022).

However, while effective on clear and detailed images, this approach struggles in real-world scenarios where fine-grained predictions are challenging (e.g., birds flying at high altitude), leading to incorrect predictions at higher levels. To address this, we propose a model that predicts across the entire taxonomy, which we believe provides greater robustness in practical applications.

**1-2) The local classifier (top-down) approach** leverages local information, such as higher-level class predictions, to make predictions at the next level. This design allows predictions at arbitrary nodes by stopping the inference process when a certain decision threshold is met, leading to more reliable predictions at higher levels (Deng et al., 2010; Wu et al., 2020; Brust & Denzler, 2019). As a result, these methods emphasize metrics such as the correctness-specificity trade-off (Valmadre, 2022). While a single model is commonly used, HiE (Jain et al., 2023) adjusts fine-level predictions post-hoc using coarse predictions from independently trained classifiers. However, a disadvantage of this top-down approach is the propagation of errors from higher-level predictions to lower levels.

**2) The full-taxonomy prediction (global classifier) approach** aims to predict the entire taxonomy *at once*, unlike prior approaches. Most popular and effective methods uses a shared backbone with separate branches for each level (Zhu & Bain, 2017; Wehrmann et al., 2018; Chang et al., 2021; Liu et al., 2022; Chen et al., 2022; Jiang et al., 2024; Zhang et al., 2024). The key difference lies in how the hierarchical relationships are modeled. For instance, in FGN (Chang et al., 2021), finer features are concatenated to predict coarse labels, whereas in HRN (Chen et al., 2022), coarse features are added to finer features through residual connections. A critical issue in this approach is maintaining *consistency* with the taxonomy in the predicted labels. To address this, Wang et al. (2023a) proposed a consistency-aware method by adjusting prediction scores through coarse-to-fine deduction and fine-to-coarse induction. However, we observed that using separate branches can lead to inconsistency, as each branch processes the image independently. To address this, we propose a model based on consistent visual grounding. To the best of our knowledge, no prior work has utilized visual segments to resolve inconsistency in hierarchical classification.

**Hierarchical Semantic Segmentation** aims to group and classify each pixel according to a class hierarchy (Li et al., 2022; Singh et al., 2022; Li et al., 2023; He et al., 2023; Wang et al., 2024; Qi et al., 2024), with pixel grouping varying based on the taxonomy used. However, these works require *pixel-level annotations*, which are not available in hierarchical classification. In addition, while these method focus on precise pixel-level grouping, our work leverage unsupervised segments of varying granularities within the image for hierarchical clasification.

**Unsupevised/Weakly-supervised Semantic Segmentation** aims to group pixels without pixel-level annotations or using only class labels (Hwang et al., 2019; Ouali et al., 2020; Ke et al., 2022; 2024). These works employ hierarchical grouping to achieve meaningful segmentation *without* pixel-level labels. Here, "hierarchical" refers to part-to-whole visual grouping, where smaller units (e.g., a person's face or arm) are grouped into larger regions (e.g., the whole body). Based on our intuition

that fine-grained classifiers need more detailed information, while coarse classifiers focus on broader groupings, our approach leverages these varying types of visual grouping. To implement this, we adopt the recently proposed CAST (Ke et al., 2024), whose graph pooling naturally supports consistent visual grouping. Notably, our work introduces the novel insight that part-to-whole segmentation can align with taxonomy hierarchies (e.g., finer segments for fine-grained labels, coarser segments for coarse labels), a connection not previously explored.

## C  BENCHMARK DATSET

Table 4: **Benchmark Datasets.**

| Datasets | L-17 | NL-26 | E-13 | E-30 | CUB | Aircraft | iNat21-Mini |
|---|---|---|---|---|---|---|---|
| # Levels | 2 | 2 | 2 | 2 | 3 | 3 | 3 |
| # of classes | 17-34 | 26-52 | 13-130 | 30-120 | 13-38-200 | 30-70-100 | 273-1,103-10,000 |
| # Train images | 44.2K | 65.7K | 167K | 154K | 5,994 | 6,667 | 500K |
| # Test images | 1.7K | 2.6K | 6.5K | 6K | 5,794 | 3,333 | 100K |

## D  HYPERPARAMETERS FOR TRAINING.

For a fair comparison, we use ViT-Small and CAST-Small models of corresponding sizes. As in CAST, we train both ViT and CAST using DeiT framework (Touvron et al., 2021), and segmentation granularity is set to 64, 32, 16, 8 after 3, 3, 3, 2 encoder blocks, respectively. Our training progresses from fine to coarse levels, with each segment corresponding accordingly. The initial number of superpixels is set to 196, and all data is trained with a batch size of 256. Following the literature (Chen et al., 2022), we use ImageNet pre-trained models for the Aircraft, CUB, and iNat datasets. For the ImageNet subset BREEDS dataset, we train the models from scratch. We show hyper-parameter settings in Table 5.

Table 5: **Hyper-parameters for training H-CAST and ViT on FGVC-Aircraft, CUB-200-2011, BREEDS, and iNaturalist datasets.** We follow mostly the same set up as CAST (Ke et al., 2024).

| Parameter | Aircraft | CUB, BREEDS, iNaturalist |
|---|---|---|
| batch_size | 256 | 256 |
| crop_size | 224 | 224 |
| learning_rate | $1e^{-3}$ | $5e^{-4}$ |
| weight_decay | 0.05 | 0.05 |
| momentum | 0.9 | 0.9 |
| total_epochs | 100 | 100 |
| warmup_epochs | 5 | 5 |
| warmup_learning_rate | $1e^{-4}$ | $1e^{-6}$ |
| optimizer | Adam | Adam |
| learning_rate_policy | Cosine decay | Cosine decay |
| augmentation (Cubuk et al., 2020) | RandAug(9, 0.5) | RandAug(9, 0.5) |
| label_smoothing (Szegedy et al., 2016) | 0.1 | 0.1 |
| mixup (Zhang et al., 2017) | 0.8 | 0.8 |
| cutmix (Yun et al., 2019) | 1.0 | 1.0 |
| $\alpha$ (weight for TK loss) | 0.5 | 0.5 |
| ViT-S: # Tokens | $[196]_{\times 11}$ | |
| CAST-S: # Tokens | $[196]_{\times 3}, [64]_{\times 3}, [32]_{\times 3}, [16]_{\times 2}$ | |

# E ADDITIONAL EXPERIMENTS

## E.1 COMPARISON BETWEEN FPA AND TICE.

FPA evaluates both accuracy and consistency, while TICE focuses solely on consistency. Achieving high FPA is the primary goal in hierarchical classification. The distinction between FPA and TICE is shown in Table 6.

Table 6: **FPA considers both correctness and consistency.** While TICE (Wang et al., 2023a) measures only consistency, FPA marks predictions as positive only when they are both correct and consistent.

| | | | | | | | | |
|---|---|---|---|---|---|---|---|---|
| TICE | ✓ | ✓ | ✗ | ✗ | ✗ | ✗ | ✓ | ✓ |
| FPA | ✓ | ✗ | ✗ | ✗ | ✗ | ✗ | ✗ | ✗ |

## E.2 VISUALIZATION OF ATTENTION MAP

To validate our claim that the model guides classifiers toward consistent visual grounding, we visualize attention maps from H-CAST in Figure 9. The visualizations demonstrate that as we progress from lower to upper blocks, the model increasingly attends to similar regions. In the lower blocks, attention is more detailed and localized, while in the upper blocks, attention expands to cover broader regions, including those highlighted by the lower blocks. These patterns align with our intended design for visual grounding in hierarchical classification.

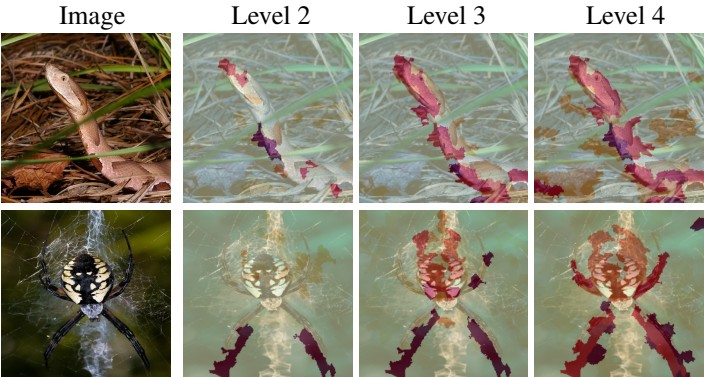

Figure 9: **Visualizations of Attention maps from H-CAST.** We align the attention weights with superpixels and average them across all heads. Darker red areas represent regions with higher attention weights. At the lower level (level 2), the attention is more focused on specific regions, such as the snake's head and parts of its body, emphasizing these as critical features for fine-grained classification (e.g., "Hypsiglena torquata"). In contrast, at the upper level (level 4), the attention expands to encompass the entire body of the snake, suggesting a shift towards a more holistic understanding of the object for coarse label (e.g., "snake"). This progression from localized to broader attention illustrates how H-CAST hierarchically integrates information across layers, supporting consistent visual grounding for hierarchical classification.

E.3 ADDITIONAL EXPERIMENTS FOR LOSS ABLATION

Similar to the results on the Aircraft dataset, the Living-17 dataset also shows consistent performance trends, with our proposed loss achieving strong results (Table 7, Table 8). Interestingly, for TICE, which measures only semantic consistency, the TK loss alone (Table 7) and the BCE or Flat Consistency loss achieved better performance (Table 8). However, when considering both accuracy and consistency (i.e., FPA), our proposed loss delivered the best overall performance.

Table 7: **Utilizing both losses yields best performance on Living-17.**

| $L_{HS}$ | $L_{TK}$ | FPA | Coarse | Fine | wAP | TICE |
|---|---|---|---|---|---|---|
| ✗ | ✓ | 84.00 | 90.71 | 84.30 | 86.43 | **1.71** |
| ✓ | ✗ | 84.21 | 90.24 | 84.59 | 86.78 | 2.59 |
| ✓ | ✓ | **85.12** | **90.82** | **85.24** | **87.10** | 3.19 |

Table 8: **KL Div. loss shows best performance on Living-17.**

| Sem. Consis. | FPA | Coarse | Fine | wAP | TICE |
|---|---|---|---|---|---|
| Flat Cons. | 82.82 | 88.88 | 83.53 | 85.31 | 2.51 |
| BCE | 83.65 | 89.76 | 84.00 | 85.92 | **1.76** |
| KL Div. | **85.12** | **90.82** | **85.24** | **87.10** | 3.19 |

E.4 EVALUATION ON THE LARGE-SCALE INATURALIST DATASETS.

We present the results of our experiments on the large-scale dataset, iNaturalist21-mini (Van Horn et al., 2021) and iNaturalist-2018 (Horn et al., 2018). First, iNaturalist21-mini contains 10,000 classes, 500,000 training samples, and 100,000 test samples, organized within an 8-level hierarchy. For our experiments, we focused on a 3-level hierarchy consisting of *name*, *family*, and *order*. This selection was motivated by the need for models with practical and meaningful granularity for real-world applications.

The number of classes at each hierarchical level is as follows: Kingdom (3), Supercategory (11), Phylum (13), Class (51), Order (273), Family (1,103), Genus (4,884), Name (10,000)

We excluded coarse-grained levels such as *kingdom* (3 classes), *Supercategory* (11 classes), because their minimal granularity adds little value for classification tasks. Similarly, overly fine-grained levels such as *genus* (4,884 classes), where many species are represented by only one or two samples, offer limited differentiation from direct *name*-level classification. Instead, we focused on *order* (273 classes), *family* (1,103 classes), and *name* (10,000 classes) to ensure that each higher-level class represents a meaningful subset of lower-level classes, allowing for interpretable and consistent predictions.

The results are shown in Table 9. Compared to ViT baselines that use the same ViT-small backbone, our method achieves a 5.98% improvement over TransHP (Wang et al., 2023b) and a 9.27% improvement over Hier-ViT in the FPA metric, demonstrating a significant performance advantage.

Table 9: **Our H-CAST outperforms ViT-backbone baselines, Hier-ViT and TransHP, on the large-scale iNaturalist2021-Mini dataset, achieving significantly higher accuracy and consistency.**

| | iNaturalist2021-Mini (273 - 1,103 - 10,000) | | | | | |
|---|---|---|---|---|---|---|
| | FPA(↑) | order(↑) | family(↑) | name(↑) | wAP(↑) | TICE(↓) |
| Hier-ViT | 55.65 | 87.25 | 77.81 | 62.71 | 64.76 | 26.21 |
| TransHP | 58.94 | 83.49 | 82.15 | 68.82 | 70.46 | 24.63 |
| Ours (H-CAST) | **64.92** | **89.72** | **84.00** | **70.00** | **71.83** | **16.00** |
| *Our Gains* | +5.98 | +2.47 | +1.18 | +1.27 | +1.37 | +8.63 |

We further compare our method with single-level approaches that utilize hierarchical labels to enhance fine-grained accuracy (*e.g.*, Guided (Garnot & Landrieu, 2020), HiMulConE (Zhang et al., 2022), and TransHP (Wang et al., 2023b)). The comparison is conducted on the large-scale iNaturalist-2018 dataset, following TransHP. iNaturalist-2018 includes two-level hierarchical annotations with 14 super-categories and 8,142 species, comprising 437,513 training images and 24,426 validation images. We use the same H-CAST-small and the model is trained for 100 epochs, using the same hyper-parameters in Table 5. As shown in Table 10, our method achieves strong fine-grained accuracy, demonstrating the effectiveness of consistent visual grounding.

Table 10: **H-CAST outperforms methods leveraging hierarchical labels for fine-grained accuracy on the large-scale iNaturalist-2018 dataset, demonstrating the effectiveness of visual consistency.** The results are reported from TransHP (Wang et al., 2023b).

|  | iNaturalist-2018 (Acc.) |
| --- | --- |
| Guided | 63.11 |
| HiMulConE | 63.46 |
| TransHP | 64.21 |
| H-CAST | **67.13** |

### E.5 EVALUATION ON CUB-200-2011 AND FGVC-AIRCRAFT DATASETS.

Table 11 and 12 presents results on CUB and Aircraft datasets. In our experimental results, we first observe a significant performance drop of Hier-ViT compared to Flat-ViT. This highlights a common challenge in hierarchical recognition, where training coarse and fine-grained classifiers simultaneously results in performance degradation, as observed in previous ResNet-based hierarchical recognition models (Chang et al., 2021). Our experiments reveal that this problem also exists in ViT architectures. This indicates that hierarchical recognition is a challenging problem that cannot be solely addressed by providing hierarchy supervision to class tokens. On the other hand, our method consistently outperforms most Flat models.

Compared to ViT-backbone models, Hier-ViT and TransHP, our approach achieves significantly better performance. Specifically, using the *FPA* metric, which captures both accuracy and consistency across all levels, our model outperforms TransHP by +8.2%p on the Aircraft dataset and +2.6%p on the CUB dataset.

We also evaluate BIOCLIP (Stevens et al., 2024), a foundation model for biology, on the CUB dataset, as it focuses on bird categories. BIOCLIP operates as a flat-based hierarchical model, concatenating the entire taxonomy into a single text representation. As a result, all higher-level classes are directly determined by the fine-grained species predictions, resulting in a TICE (Taxonomy-Inconsistency Error) of 0. While BIOCLIP achieves strong performance, its reliance on fine-grained predictions to define coarse classes introduces limitations in accurately predicting higher-level classes.

As Vision Transformer backbone models, when the training dataset is small, such as Aircraft and CUB with around 6K images, HRN, ResNet-based models, demonstrates better performance. However, HRN's method is highly sensitive to batch size, with a significant drop in performance observed when increasing the batch size from 8 to 64. This sensitivity makes it less suitable for training on large-scale datasets.

Table 11: **Ours consistently shows the best performance on CUB-200-2011.** H-CAST outperforms ViT-backbone models, Hier-ViT and TransHP, by over 6.3 and 2.6 percentage points, respectively. Additionally, it achieves a 3.2 percentage point gain in the FPA metric over the ResNet-based HRN while using significantly fewer parameters. (A higher metric indicates better performance, except for TICE.) Flat models require training three separate models.

| | | Configuration | | | CUB-200-2011 (13 - 38 - 200) | | | | | |
|---|---|---|---|---|---|---|---|---|---|---|
| | | backbone | #params | input | batch size | FPA | order | family | species | wAP | TICE |
| Flat | Flat-ViT | ViT-S | 65.1M | $224^2$ | 256 | 82.30 | 98.50 | 94.84 | 84.78 | 87.01 | 5.76 |
| | Flat-CAST | ViT-S | 78.5M | $224^2$ | 256 | 81.50 | 98.38 | 94.82 | 83.78 | 86.21 | 6.14 |
| Hierarchy | FGN | RN-50 | 24.8M | $224^2$ | 128 | 76.08 | 97.05 | 91.44 | 79.29 | 82.05 | 7.73 |
| | HRN | RN-50 | 94.5M | $448^2$ | 64 | 80.07 | 98.17 | 93.75 | 83.14 | 85.52 | 6.51 |
| | HRN | RN-50 | 94.5M | $448^2$ | 8 | **84.15** | 98.58 | **95.39** | **86.13** | **88.18** | 4.62 |
| | BIOCLIP (zeroshot) | ViT-B | 149.6M | - | - | 78.18 | 78.18 | 78.18 | 78.18 | 78.18 | **0.0** |
| | Hier-ViT | ViT-S | 21.7M | $224^2$ | 256 | 77.03 | 98.40 | 92.94 | 79.43 | 82.46 | 8.72 |
| | TransHP | ViT-S | 21.7M | $224^2$ | 128 | 80.70 | 96.70 | 94.15 | 84.59 | 86.66 | 7.16 |
| | Ours (H-CAST) | ViT-S | 26.2M | $224^2$ | 256 | 83.28 | **98.65** | 95.12 | 84.86 | 87.13 | **4.12** |
| | *Our Gains over SOTA* | | | | | -0.87 | +0.07 | -0.27 | -1.27 | -1.05 | +0.50 |

Table 12: **Evaluation on FGVC-Aircraft.** On the smaller Aircraft dataset, ResNet-based models such as FGN and HRN show good performance. However, our H-CAST achieves better results in the consistency metric (TICE) and performs comparably in the FPA metric. Notably, H-CAST outperforms ViT-backbone models, Hier-ViT and TransHP, by over 11 and 8 percentage points, respectively, in the FPA metric.

| | | Configuration | | | FGVC-Aircraft (30 - 70 - 100) | | | | | |
|---|---|---|---|---|---|---|---|---|---|---|
| | | backbone | #params | input | batch size | FPA | maker | family | model | wAP | TICE |
| Flat | Flat-ViT | ViT-S | 65.1M | $224^2$ | 256 | 76.99 | 94.27 | 91.93 | 80.14 | 86.39 | 10.98 |
| | Flat-CAST | ViT-S | 78.5M | $224^2$ | 256 | 78.22 | 92.95 | 88.93 | 82.39 | 86.26 | 10.77 |
| Hierarchy | FGN | RN-50 | 24.8M | $224^2$ | 128 | 85.48 | 92.44 | 90.88 | 88.39 | 89.87 | 7.50 |
| | HRN | RN-50 | 94.5M | $448^2$ | 64 | 83.56 | 94.93 | 92.68 | 86.59 | 89.97 | 7.26 |
| | HRN | RN-50 | 94.5M | $448^2$ | 8 | **91.39** | **97.15** | **95.65** | **92.32** | **94.21** | **3.36** |
| | Hier-ViT | ViT-S | 21.7M | $224^2$ | 256 | 72.10 | 92.35 | 86.26 | 75.94 | 82.01 | 15.75 |
| | TransHP | ViT-S | 21.7M | $224^2$ | 128 | 75.49 | 90.16 | 87.46 | 81.46 | 84.86 | 13.95 |
| | Ours (H-CAST) | ViT-S | 26.2M | $224^2$ | 256 | 83.72 | 94.96 | 91.39 | 85.33 | 88.90 | 5.01 |
| | *Our Gains over SOTA* | | | | | -7.67 | -2.18 | -4.26 | -6.99 | -5.31 | -1.65 |

## E.6 ADDITIONAL VISUALIZATIONS OF SEGMENTS

We visualize additional examples of feature grouping from fine to coarse for full-path correct and incorrect predictions on the Entity-30 dataset in Figure 10. For full-path correct predictions (all levels correct), visual details are effectively grouped to identify larger objects at coarser levels. In contrast, for full-path incorrect predictions (all levels incorrect), segments fail to recognize the object.

Fine to coarse segments          Fine to coarse segments

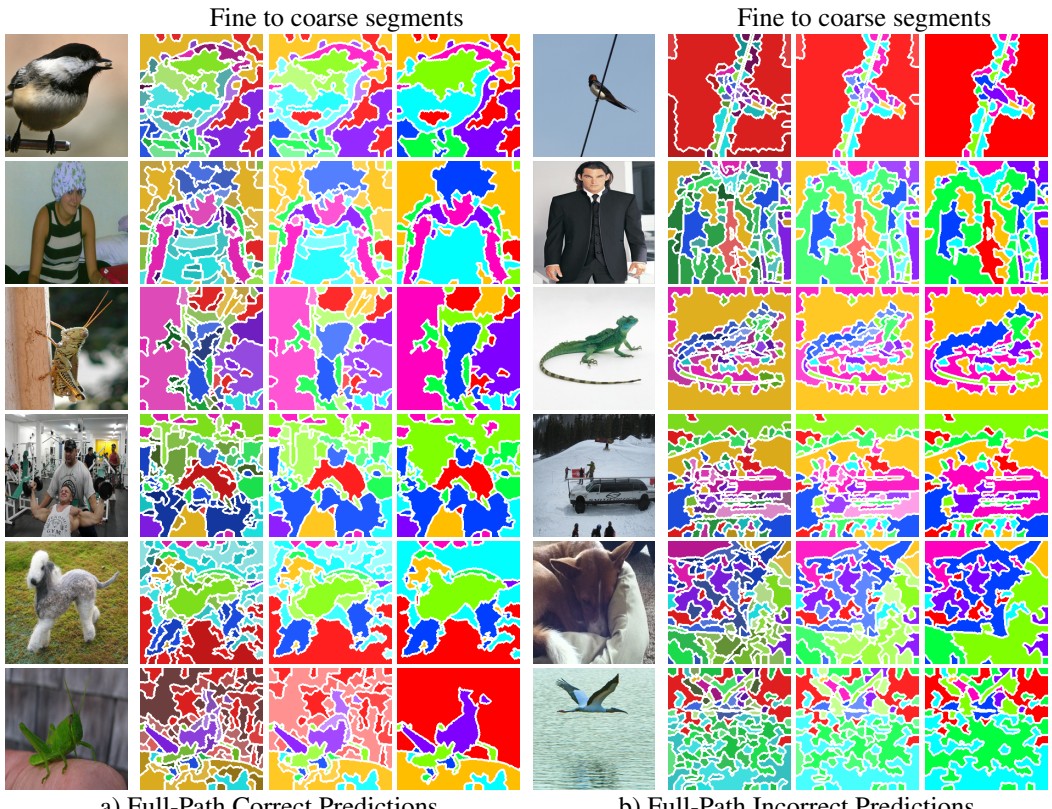

a) Full-Path Correct Predictions          b) Full-Path Incorrect Predictions

Figure 10: Additional examples of the differences in visual grouping between cases where predictions at all levels are correct and where they are not. Correct predictions show better clustering, while incorrect predictions often exhibit fractured or misaligned groupings.

### E.7    COMPARISON OF IMAGE SEGMENTATION WITH CAST

To quantitatively evaluate the segmentation results in Figure 8, we use the ImageNet segmentation dataset, ImageNet-S (Gao et al., 2022), to obtain the ground-truth segmentation data for BREEDS dataset. The number of samples in the BREEDS validation data for which ground-truth segmentation data can be obtained from ImageNet-S is 381 for Living-17, 510 for Non-Living-26, 1,336 for Entity-30, and 1,463 for Entity-13. To calculate the region mIOU for fine-level objects, we use the last-level segments (8-way) for segmentation. Following CAST, we name the 8-way segmentations using OvSEG (Liang et al., 2023).

Also, we further visualize the segmentation results on Entity-30 in Figure 11, and show that additional taxonomy information improves segmentation. For example in the first 'bird' image, H-CAST is able to segment meaningful parts such as the face, belly, and a branch, with less fractured compared to the CAST. Thus, H-CAST delivers an improvement in segmentation with the benefits of hierarchy.

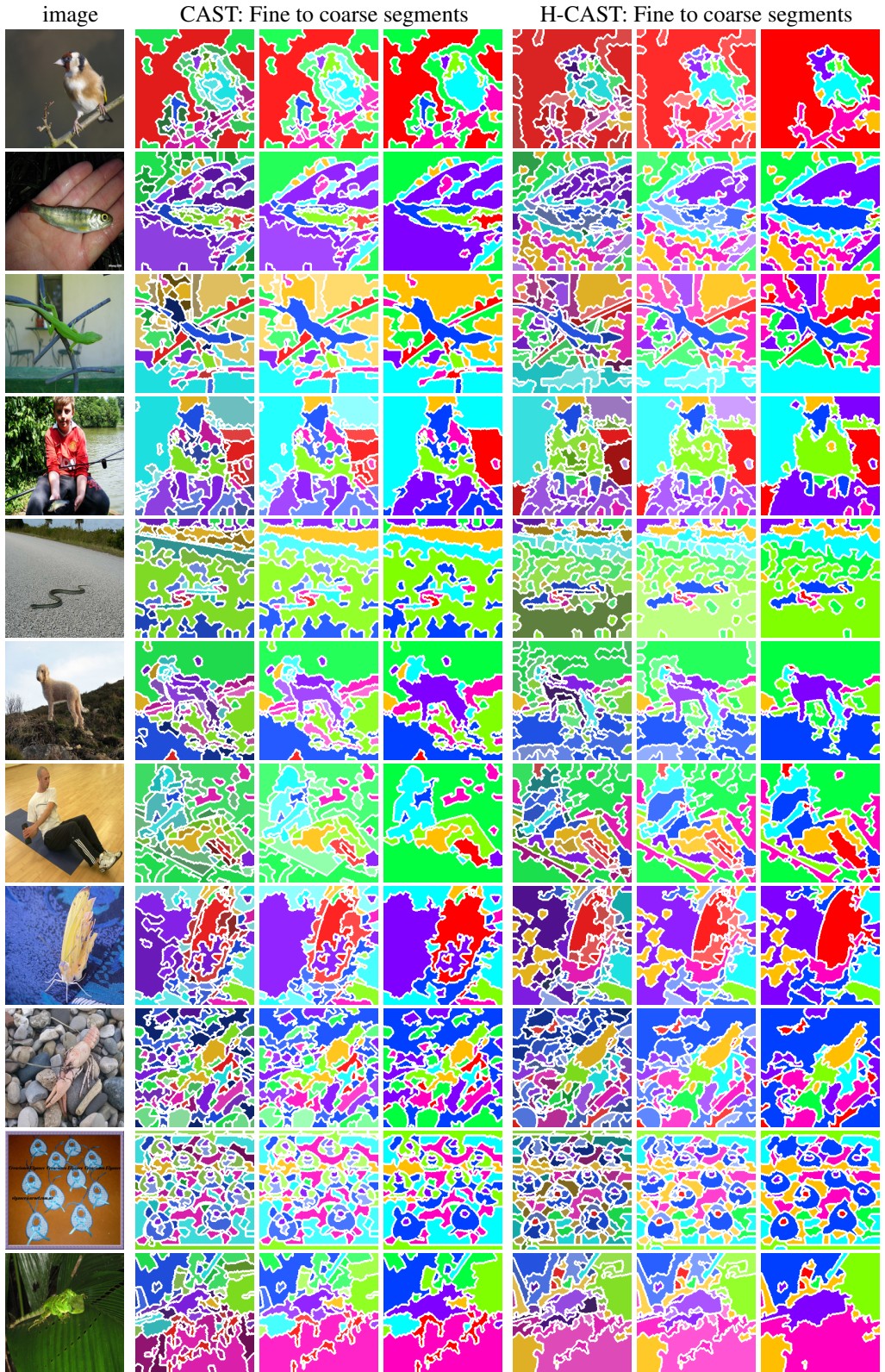

Figure 11: Additional visual results on segmentation show that H-CAST with additional taxonomy information improves segmentation. H-CAST successfully segments meaningful parts with fewer fractures compared to CAST.

