# OpenReview forum: "Visually Consistent Hierarchical Image Classification"
_ICLR.cc/2025/Conference — ICLR 2025 Poster_

### Official Review · Reviewer_uxN6 · 2024-10-28

**Soundness:** 2
**Presentation:** 3
**Contribution:** 2
**Rating:** 3
**Confidence:** 5

**Summary:**

The paper explores hierarchical image classification (HIC) task, which is old but worthy to explore. I have experienced many bad reviews in this domain, for instance, “the proposed method relies on coarse labels and therefore not useful in the real world”. However, I think they are the misunderstanding of this area and I want to give some new one.

**Strengths:**

NA

**Weaknesses:**

Overlap with former work? I noticed that an important reference, TransHP: Image Classification with Hierarchical Prompting (NeurIPS 2023), is not cited in your paper. This work may be directly relevant, as it appears to share similarities with your proposed method, specifically in the use of different ViT blocks for different levels of hierarchy.

Limited novelty. Your proposed approach introduces elements such as Superpixel and Graph pooling. While these are effective, both are well-established techniques in computer vision. A more detailed explanation of how these additions provide novel contributions within the hierarchical framework would clarify the unique aspects of your work.

Limited evaluation. ImageNet, as a large-scale hierarchical dataset, might provide a stronger test of your method's capabilities compared to the smaller datasets currently used.

Formatting problem: Please ensure the readability of all components of the paper. For instance, the font size in the tables is small, which may make it difficult for readers to check the data presented

**Questions:**

Please specific what is the difference between your work and TransHP?

**Details Of Ethics Concerns:**

I doubt the contribution of this paper beyond a published one [1] in NeurIPS 2023. The paper fails to cite it and may intend to mislead the readers.

[1] TransHP: Image Classification with Hierarchical Prompting

---

> ### Author Response · Authors · 2024-11-22
>
> Dear reviewer uxN6,
> Thank you for your valuable feedback and comments. We address your concerns and questions in the response below.
>
>
> **1. Overlap with TransHP (NeurIPS 2023)**
> >1. Overlap with former work? I noticed that an important reference, TransHP: Image Classification with Hierarchical Prompting (NeurIPS 2023), is not cited in your paper. ....
>
>
>
> We would like to clarify the differences between our work and TransHP to address the concern. While both works use ViT backbones, our approach and TransHP differ fundamentally in both goals and methodology:
>
> 1. **Goal**:
>    TransHP focuses on improving fine-grained classification by leveraging hierarchical information as auxiliary supervision. In contrast, our work targets multi-granularity classification, aiming to make predictions across all levels of the hierarchy simultaneously while ensuring semantic consistency between levels.
>
> 2. **Method**:
>    The approach to coarse-level supervision is also significantly different. TransHP uses learnable prompts, with one prompt per coarse label, to guide fine-grained predictions. On the other hand, we directly apply coarse-level supervision to the output class tokens of each block, enabling independent predictions at different levels and facilitating consistent visual grounding across the hierarchy.
>
> These differences demonstrate that our work addresses a distinct challenge with a fundamentally different methodology. We included TransHP in the related work of the revised manuscript and clarify these distinctions to avoid any potential confusion.
>
>
> ----
> **2. Limited novelty**
> > Limited novelty. Your proposed approach introduces elements such as Superpixel and Graph pooling. While these are effective, both are well-established techniques in computer vision....
>
> The use of superpixels and graph pooling is not part of our proposed contributions. As stated in L183-185 of the original manuscript, these are components introduced by CAST, and we clearly attribute them as CAST's innovations.
>
> Our novel contributions are as follows:
>
>
> (1) **Key Insight**:
>   Our observation revealed that classification at different granularities involves fundamentally distinct tasks requiring attention to different but consistent regions within an image. We found that inconsistencies arise because each classifier tends to independently attend to different regions without connection. This observation led us to propose consistent visual grounding as a novel solution to connect hierarchical classifiers across levels.
>
> (2) **Leveraging Semantic Segments for Hierarchical Classification**:
>    While we adopted CAST as part of our architecture, it is important to emphasize that CAST originates from a different task, weakly-supervised semantic segmentation. In CAST, “hierarchy” refers to “part-to-whole” visual grouping (e.g., eyes, nose, arms), while our work addresses a “taxonomy hierarchy” (e.g., bird - Green Hermit). It was NOT evident that the concept of “part-to-whole” segments would align well with a taxonomy hierarchy; this connection is a novel discovery introduced through our work.
>
> In addition, based on our observation in (1), we newly propose leveraging segments at different granularities to enhance multi-granularity classification. To the best of our knowledge, the use of segments has NOT been applied to hierarchical classification tasks.
>
> Thus, this adaptation is neither trivial nor an obvious solution; it stems from our novel observation and bridges two distinct fields to tackle challenges unique to hierarchical classification.
>
> We hope this summary highlights the novelty and importance of our work.

---

> ### Author Response · Authors · 2024-11-22
> **Official Comment by Authors (2)**
>
> **3. Limited Evaluation on ImageNet**
> > Limited evaluation. ImageNet, as a large-scale hierarchical dataset, might provide a stronger test of your method's capabilities compared to the smaller datasets currently used.
>
> We appreciate the reviewer’s comments and would like to clarify our dataset choices and experimental setup.
>
> First, **CUB and Aircraft are the most commonly used benchmark datasets in prior studies [1, 2, 3, 4]**, and we conducted our experiments on these datasets to ensure comparability with existing methods. Additionally, to evaluate our approach on a larger and more challenging subset that includes diverse classes beyond a single type (e.g., birds or aircraft), we conducted experiments on **BREEDS, a subset of ImageNet**.
>
> Regarding ImageNet-1K, its hierarchy is highly imbalanced, making it challenging to apply our approach. As a result, most hierarchical classification studies on this dataset have focused on **flat-level classification**, not multi-granularity classification.
>
> We are currently conducting additional experiments on  large-scale iNaturalist dataset to further validate our method on a larger dataset. We will update the results as they become available. However, we kindly ask for your understanding, as our training environment relies on a single GPU (Nvidia A40), which causes the experiments to take longer to complete.
>
> [1] Your “Flamingo” is My “Bird”: Fine-Grained, or Not, 2021
> [2] Consistency-aware feature learning for hierarchical fine-grained visual classification, 2023
> [3]  Hierarchical multi-granularity classification based on bidirectional knowledge transfer, 2024
> [4] HLS-FGVC: Hierarchical Label Semantics Enhanced Fine-Grained Visual Classification, 2024
>
> ----
> **4. Formatting problem**
> >Formatting problem: Please ensure the readability of all components of the paper. For instance, the font size in the tables is small, which may make it difficult for readers to check the data presented
>
>
> Thank you for pointing out the formatting issue. Due to the page limit, we had to make certain adjustments to fit the content. However, we revised Table 2 to increase its font size for better readability. If there are specific tables or components that are still difficult to view, please let us know, and we will address them further.
>
>
> ---
> We hope this clarifies the reviewer’s concerns. If there are any further concerns or clarifications needed, we would be happy to discuss them further.

---

> > ### Comment · Reviewer_uxN6 · 2024-11-25
> >
> > Dear authors,
> > Thanks very much for your reply. However, my concerns remain unsolved. Therefore, I maintain my 3 score. The details are as follows:
> >
> > 1. The difference between TransHP:
> >
> > You argue two points about differences:
> >
> > (1) Goal: you said the goals between yours and TransHP are different. However, from my point of view, they are similar. TransHP uses *hierarchical prompting* to improve the performance of finer classification. It does not constrain to the last layer, and it can also easily be adapted to multi-granularity classification as yours.
> >
> > (2) Method: the difference argued by you is not true, I think. What you argue is different is just a variant of TransHP. TransHP also conducts ablation studies on this variant: see Fig. 4 (2). That is exactly the same with yours.
> >
> > 2. Given 1, the novelty is limited.
> >
> > 3. The lack of GPU should not be an excuse for not performing experiments on large-scale datasets. They can be easily accessed on the Cloud.
> >
> > 4.  Thanks for updating the figures.

---

> > ### Comment · Reviewer_uxN6 · 2024-11-25
> >
> > By the way, I think this paper has content laundering of TransHP. However, because I do not specialize in copyright/plagiarism, I cannot draw any conclusions. And there is no ethnic reviewer for this paper.

---

> ### Author Response · Authors · 2024-11-25
>
> We appreciate the reviewer’s feedback, and we would like to provide further clarification.
>
> **1. The difference between TransHP**
>
> (1) TransHP’s variant in Fig.4 (2) is NOT "*exactly the same*" as ours. Instead, it can be regarded as  the **Hier-ViT** baseline, which is among the baselines we employed for comparison.
>
> (2) While both our method and TransHP utilize a ViT backbone and incorporate coarse labels, the similarities are limited to these general design choices. The fundamental differences lie in the motivation, the role of coarse labels, and the methodology to hierarchical classification.
>
> To clarify, hierarchical classification poses a unique challenge: while the input image remains constant, the output shifts across the semantic (text) space (e.g., from "bird" to "Green Hermit"). Due to this formulation, existing works, including TransHP, primarily address this challenge by embedding data into the **semantic space**.
>
> Specifically, **TransHP** employs coarse labels as prompts to guide fine-grained classification within the hierarchy, operating primarily in the **semantic space**. The focus is on **refining predictions within the scope of coarse label prompts**, embedding hierarchical information into text-based representations.
>
> In contrast, our approach is fundamentally distinct, as it addresses hierarchical classification through the **visual space**. Rather than relying on hierarchical labels as prompts, we investigate how hierarchical classification can be linked to visual grounding, analyzing images at varying levels of detail—ranging from fine-grained to holistic representations. Additionally, we **specifically linked detailed part-level segments to fine-grained labels and coarse segments to coarse labels, ensuring that each level of unsupervised visual segments contributes effectively to its corresponding learning process**. This emphasis on **visual grounding** forms the core of our contribution, setting it apart from TransHP.
>
> Additionally, our supervision strategy further differentiates the two approaches. While TransHP adopts a **coarse-to-fine supervision** framework, we reverse this by applying supervision from **fine-grained to coarse labels** from our *consistent visual grounding motivation*. This novel formulation enables a stronger and more consistent alignment between visual features and hierarchical levels, further underscoring the originality of our work.
>
> In summary, while both methods incorporate coarse labels and share a ViT backbone, our focus on **visual grounding** and the **fine-to-coarse supervision paradigm** highlights a fundamentally different and innovative approach to hierarchical classification.
>
> ----------------
>
> **2. Experiments on larger dataset, iNat21-Mini**
>
> We want to make it clear that the lack of GPU resources was NOT used as an excuse to avoid conducting experiments. Rather, we simply requested additional time due to the constraints involved. The updated results for the iNaturalist dataset are now provided below.
>
> It is also worth emphasizing that not all research environments have access to abundant computational resources, such as 8 A100 GPUs in the Cloud. Research conducted with limited GPU resources, while focusing on diverse experiments on smaller datasets, is no less meaningful and can provide valuable insights. We stand by the validity and contribution of our approach under these circumstances.
>
> We'd like to share the results of our experiments on the large-scale dataset (iNaturalist 2021-mini). iNat21-mini [1] contains a total of **10,000 classes** and **500,000 training samples** and **100,000 test samples**. For our experiments, we focused on a 3-level hierarchy consisting of  **order** (**273** classes), **family** (**1,103** classes), and **name** (**10,000** classes) for our 3-level hierarchy.
>
> The results are presented in the table below. Compared to Hier-ViT, which uses the same ViT-small backbone, our method demonstrates the **fine-grained accuracy improvement of over 7.29%** and a **8.3% improvement in the FPA metric**, representing a significant performance gain.
>
> We hope these results address concerns about large-scale data.
> We will include this experimental result in the revised manuscript.
>
> **< iNat21-Mini (273 - 1,103 - 10,000) >**
> |  | FPA | Order | Family | Name | wAP | TICE |
> |---|:---:|:---:|:---:|:---:|:---:|:---:|
> | Hier-ViT | 56.73 | 87.54 | 79.79 | 62.81 | 65.05 | 24.34 |
> | Ours | **65.03** | **89.84** | **84.12** | **70.09** | **71.92** | **15.92** |
>
>
> [1]  Benchmarking representation learning for natural world image collections. 2021

---

> > ### Comment · Reviewer_uxN6 · 2024-11-26
> >
> > Thanks for your reply. However I insist my opinion that this paper has limited novelty. Similarly, Reviewers ZNGF and khL4 have raised the same concerns. This paper is apparently below the bar of ICLR.

---

### Official Review · Reviewer_ZNGF · 2024-10-30

**Soundness:** 2
**Presentation:** 2
**Contribution:** 2
**Rating:** 5
**Confidence:** 5

**Summary:**

This work aims to tackle hierarchical image classification. It is motivated by a hierarchical image segmentation work, which also conducts image classification. This work makes an extension on this basis and proposes a tree KL loss to deliver semantic consistency predictions regarding hierarchy. Evaluation on three datasets verifies the effectiveness over the baseline, which focuses on segmentation rather than classification.

**Strengths:**

1. The overall presentation is easy to follow. There is no difficulty in understanding the work.
2. The improvement on the baseline is considerable.

**Weaknesses:**

1. The technical contribution is not enough. Specifically, the proposed method contains visual consistency and semantic consistency. However, the major design and implementation of visual consistency are directly borrowed from CAST [1]. The claim in L260-264 is not convincing. Simply adding supervision in each decoding level can not be considered a vital contribution compared to directly using the overall architecture of CAST.

[1]. Learning hierarchical image segmentation for recognition and by recognition. ICLR 2024

2. The review only contains hierarchical image classification. However, pixel-level hierarchical classification (i.e., hierarchical image segmentation [2,3,4]) can also provide insights for this work. In fact, the visual consistency mentioned in this work is conducting segmentation on the image, and the baseline method (CAST) is also a hierarchical image segmentation work. Therefore, a literature review on hierarchical image segmentation should be included.

[2] AIMS: All-Inclusive Multi-Level Segmentation for Anything. NeurIPS 2023.

[3] Hierarchical Open-vocabulary Universal Image Segmentation. NeurIPS 2023

[4] LOGICSEG: parsing visual semantics with neural logic learning and reasoning. ICCV 2023.


3. Concatenate labels from all levels to create a distribution has already been explored in [3,4]. The difference is [3,4] using cross-entropy loss, while this work uses KL divergence.

4. The motivation for using KL divergence instead of cross-entropy is unclear. Since CE loss contains both the KL term to minimize the differences between distribution and a punishment term to minimize the uncertainty. Why KL divergence is better than CE? Both experiments and the theoretical explanation should be provided.

5. The evaluation of the proposed method is limited.

      i) The comparison to hierarchical counterparts only contains out-of-date methods published before 2022 and the baseline (focusing on segmentation rather than classification). The top-leading solutions (e.g., [5]) are all ignored.

      ii) The label hierarchy is shallow, with only up to 3 hierarchical levels.

[5]. TransHP: Image Classification with Hierarchical Prompting. NeurIPS 2023

6. The claims in 4.6 are similar to the insights provided in [1].

Overall, the technical contribution, evaluation, and insights provided by this work are all limited.

**Questions:**

Why KL divergence is better than CE? Both experiments and the theoretical explanation should be provided.

---

> ### Author Response · Authors · 2024-11-22
>
> Dear reviewer ZNGF ,
> Thank you for your valuable feedback and comments. We appreciate your recognition of the clarity of our presentation and the significant improvements over the baseline. We address your concerns and questions in the response below.
>
>
> **1. Novel Contribution over CAST**
> > The technical contribution is not enough. Specifically, the proposed method contains visual consistency and semantic consistency. However, the major design and implementation of visual consistency are directly borrowed from CAST [1]....
>
> We thank the reviewer for the opportunity to clarify our contributions to hierarchical classification and the novel role of CAST in our work.
>
> (1) **Key Insight**:
>   Our observation revealed that classification at different granularities involves fundamentally distinct tasks requiring attention to different but consistent regions within an image. We found that inconsistencies arise because each classifier tends to independently attend to different regions without connection. This observation led us to propose consistent visual grounding as a novel solution to connect hierarchical classifiers across levels.
>
> (2) **Leveraging Semantic Segments for Hierarchical Classification**:
>    While we adopted CAST as part of our architecture, it is important to emphasize that CAST originates from a different task, weakly-supervised semantic segmentation. In CAST, “hierarchy” refers to “part-to-whole” visual grouping (e.g., eyes, nose, arms), while our work addresses a “taxonomy hierarchy” (e.g., bird - Green Hermit). It was NOT evident that the concept of “part-to-whole” segments would align well with a taxonomy hierarchy; this connection is a novel discovery introduced through our work.
>
> In addition, based on our observation in (1), we newly propose leveraging segments at different granularities to enhance multi-granularity classification. To the best of our knowledge, the use of segments has NOT been applied to hierarchical classification tasks.
>
> Thus, this adaptation is neither trivial nor an obvious solution; it stems from our novel observation and bridges two distinct fields to tackle challenges unique to hierarchical classification.
>
> We hope this summary highlights the novelty and importance of our work.
>
> ----
>
> **2. Related work on hierarchical segmentation**
> > The review only contains hierarchical image classification. However, pixel-level hierarchical classification (i.e., hierarchical image segmentation [2,3,4]) can also provide insights for this work. ...
>
> Thank you for introducing the related work. We have revised the related work section and included discussions on hierarchical semantic segmentation to provide a more comprehensive overview.
>
> ---
>
> **3. Concatenating labels in Tree-path KL divergence loss**
> >  Concatenate labels from all levels to create a distribution has already been explored in [3,4]. The difference is [3,4] using cross-entropy loss, while this work uses KL divergence.
>
> While [3,4] concatenate labels from all levels, our method differs  fundamentally in both motivation and application. To clarify, hierarchical segmentation and hierarchical classification address entirely different challenges:
>
> **Hierarchical segmentation** focuses on **spatial granularity**, identifying visual elements at different scales within the image (e.g., parts like "eye" or "head" versus the whole "person").
>
> **Hierarchical classification**, on the other hand, deals with **semantic granularity**, where the image remains the same, but the interpretation of its content varies by level (e.g., "bird" → "hummingbird" → "green hermit"). A key challenge in hierarchical classification is addressing **inconsistencies** in predictions across levels (e.g., the coarse-level classifier predicts "plant," while the fine-level classifier predicts "bird").
>
> For example, in HIPIE [3], instance class names (e.g., "person," "cat") are concatenated with part class names (e.g., "head," "eye") to capture a **compositional hierarchy** in spatial segmentation. In contrast, our method models **semantic relationships** as a distribution by one-hot encoding hierarchical labels and concatenating them.
>
> Our primary goal is to ensure **semantic consistency across levels** in hierarchical classification. To achieve this, we introduce the Tree-path KL Divergence loss, which transforms hierarchical labels into a distribution and enforces alignment across the taxonomy. This motivation fundamentally differs from [3,4], by explicitly modeling and preserving the semantic relationships between hierarchical levels.
>
> ---

---

> ### Author Response · Authors · 2024-11-22
> **Official Comment by Authors (2)**
>
> **4. Cross-entropy over KL divergence loss**
> > The motivation for using KL divergence instead of cross-entropy is unclear....   Both experiments and the theoretical explanation should be provided.
>
> The motivation for using KL divergence instead of cross-entropy lies in the need to encode semantic consistency across hierarchical levels. KL divergence enables us to model the entire hierarchical tree as a single distribution, capturing the relationships across all levels simultaneously, rather than treating each level as an independent classification task, as is common with cross-entropy loss.
> To evaluate the effectiveness of our Tree-path KL Divergence loss, we conducted experiments comparing it to two alternatives:
> - Binary Cross Entropy (BCE) loss: A widely used approach for hierarchical classification when treated as a multi-label classification task.
> - Flat Consistency loss: Inspired by bottom-up approaches, it infers coarse-level predictions from fine-grained ones and applies BCE to align them with the ground truth.
>
> As shown in the table below, Tree-path KL Divergence loss outperforms both alternatives, achieving the highest FPA on the Living-17 dataset and demonstrating superior accuracy and semantic consistency. Similar trends are observed on the  Aircraft dataset.
>
> **<Living-17 dataset>**
>
> | Semantic consistency loss | FPA | Coarse | Fine | wAP |
> |---|---|---|---|---|
> | Flat consistency loss | 82.82 | 88.88 | 83.53 | 85.31 |
> | BCE loss | 83.65 | 89.76 | 84.00 | 85.92 |
> | KL Divergence loss | 85.12 | 90.82 | 85.24 | 87.10 |
>
> **<Aircraft dataset>**
>
> | Semantic consistency loss | FPA | maker | family | model | wAP |
> |---|---|---|---|---|---|
> | Flat consistency loss | 82.87 | 94.63 | 90.94 | 84.97 | 88.51 |
> | BCE loss | 82.18 | 94.21 | 90.13 | 84.88 | 88.11 |
> | KL Divergence loss | 83.72 | 94.96 | 91.39 | 85.33 |  |
>
> We have included these results and explanations in the updated manuscript  Table 5 and 10.
>
>
> ----------
>
> **5-1 Comparison to Hierarchical Counterparts and more recent works**
> >  i) The comparison to hierarchical counterparts only contains out-of-date methods published before 2022 and the baseline (focusing on segmentation rather than classification). The top-leading solutions (e.g., [5]) are all ignored.
>
> Our evaluation includes comparisons with relevant hierarchical classification methods such as FGN and HRN, which are widely recognized benchmarks in hierarchical multi-granularity classification. Additionally, we incorporated HIE (NeurIPS 2023), a more recent method, to provide an updated comparison.
>
> Among the various directions in hierarchical classification, our work focuses on multi-granularity classification, where predictions are made simultaneously across multiple levels. In contrast, many existing methods (e.g., [5]) focus on flat classification, using coarse labels to enhance fine-grained predictions. Consequently, there are few methods available for direct comparison in multi-granularity classification.
>
> In addition, recent works on multi-granularity classification [a, b, c] have not made their code publicly available, which is why we compared against FGN and HRN. Although the goal of TransHP [5] differs from ours, we attempted to evaluate it for comparison. However, [5] requires significant resources, such as training on 8 A100 GPUs, while our server is limited to a single A40 GPU, making it challenging to reproduce their results during this rebuttal period. We kindly ask for your understanding regarding this limitation.
>
> **5-2. focusing on segmentation?:**
> We included CAST in our comparisons because we adopted it for visual grounding to address inconsistent predictions in hierarchical classification. We used it as one of the flat-level baseline. Additionally, since our work utilizes unsupervised segments to enhance hierarchical classification, it naturally raises the research question of whether this approach could also benefit segmentation in reverse. To explore this, we conducted additional experiments to evaluate its potential impact on segmentation tasks.
>
> **5-3. Shallow hierarchy**
>
> We acknowledge that the label hierarchy in our work is relatively shallow, with up to 3 hierarchical levels. However, this follows the standard practice in multi-granularity classification tasks, as seen in prior works [a, b, c].
> We acknowledge the need for scalability in deeper trees and view this as a promising direction for future research. Exploring adjustments to the loss function and other architectural adaptations to handle larger hierarchies is an exciting area we plan to investigate further.
>
> [a] Consistency-aware feature learning for hierarchical fine-grained visual classification, 2023
> [b]  Hierarchical multi-granularity classification based on bidirectional knowledge transfer, 2024
> [c] HLS-FGVC: Hierarchical Label Semantics Enhanced Fine-Grained Visual Classification, 2024

---

> ### Author Response · Authors · 2024-11-22
> **Official Comment by Authors (3)**
>
> **6. Claim in 4.6**
> > The claims in 4.6 are similar to the insights provided in [1].
>
> What we highlighted in Section 4.6 is the finding that taxonomy class supervision can be beneficial for segmentation. In CAST, the hierarchy refers to part-to-whole segments, and it was unexpected that taxonomy hierarchy could improve this. If our explanation is unclear, we would appreciate it if you could elaborate on what you mean by "similar insights" in [1], so we can address it more effectively.
>
> ----
>
> **Limited contribution**
> > Overall, the technical contribution, evaluation, and insights provided by this work are all limited.
>
> We have addressed our novel contribution in 1.
> To further clarify and strengthen our contributions, we have made several updates to the paper:
>
> 1. **Related Work Revision**: We revised the related work section to clearly position our work within the context of multi-granularity classification and to distinguish our direction from related research areas.
>
> 2. **Support for Motivation**: To reinforce our motivation, we conducted a Grad-CAM analysis, demonstrating the relationship between consistent visual grounding and improved hierarchical classification.
>
> 3. **Additional Experiments**: We added attention map visualizations to provide further interpretability of the model’s predictions. Additionally, we included a loss ablation study to evaluate the effectiveness of our Tree-path KL Divergence loss compared to other commonly used loss functions.
>
>
> We believe these updates provide a more comprehensive understanding of our technical contributions, evaluation, and insights.
>
>
> ------------
>
> We hope this clarifies the reviewer’s concerns. If there are any further concerns or clarifications needed, we would be happy to discuss them further.

---

> > ### Comment · Reviewer_ZNGF · 2024-11-26
> >
> > Thanks to the authors for their response, which has addressed some of my concerns. As a result, I have decided to raise my score to 5. However, I remain concerned about the novelty and evaluation of the proposed method.

---

> > > ### Author Response · Authors · 2024-11-26
> > >
> > > We’re pleased to hear that some of your concerns have been addressed, and we sincerely appreciate your decision to raise the score.

---

### Official Review · Reviewer_YARY · 2024-10-30

**Soundness:** 4
**Presentation:** 4
**Contribution:** 3
**Rating:** 8
**Confidence:** 4

**Summary:**

This article deals with hierarchical image classification using neural networks. This classification is actually composed of several classifications performed at different levels of precision (coarse to fine). It is important that the classes detected for an image at different levels match the logical organization of the labels. Labels are organized in a tree structure, where each level of precision is a level of the tree. Therefore, there must be a direct path between the different classes identified in an image for the classification to be correct.

So we understand that in hierarchical classification, there are different levels at which to judge the quality of the classification. At each level, there's the accuracy with respect to the expected label, and there's the logical connection between classifications at different levels.

A simple solution would be to use a single encoder to represent the image and different classification heads for each hierarchical level. The article explains that this is a bad solution because it puts the different hierarchical levels in competition with each other because they require different levels of representation. The current state of the art divides the architecture into independent branches, each of which generates a representation for a different hierarchical level. The authors have found that this separation has led to a non-causality in the classification of the different levels (decisions are independent), which could lead to inconsistencies on the relationship between classifications of different levels.


The article focuses on CAST, a variant of the transformer network that does not slice the image into patches but into superpixels and integrates a hierarchical structure by including superpixel merging. The architecture thus goes from the fine to the coarse level. Thus, the authors have found that this architecture is adapted to the problem of hierarchical classification by adopting a fine-to-coarse classification logic. Therefore, they have adopted the same architecture, but instead of classifying the different levels in parallel like the state-of-the-art, they classify them sequentially. Thus, the super pixel embedding and the class token pass through this architecture, and in the course of this processing, the class token is classified several times. The classifier heads follow each other and classify at increasingly coarse levels.

The authors also propose to combine two losses, one which is the cross entropy of the independent classification at each level, and one which is the concatenation of the probability of all classes (renormalized) where several classes are expected (one for each level). Therefore, the losses promote a local and a global level of good classification.

In the experimental section, the authors show that their sequential classification allows a good improvement in the collection of metrics (local and structural) compared to architectures that process the task in parallel. In addition, parallel architectures are larger because they have to divide the network into as many branches as there are hierarchical levels. The authors show that a smaller network can outperform a larger one if it is designed to match the structure of the problem to be solved.

The authors also added an ablation study to investigate the advantage of fine-to-coarse over coarse-to-fine, and the influence of both local and structural loss. The authors also show that hierarchical training improves segmentation results.

**Strengths:**

The paper is well written and organized. The authors have made a good effort to put the problem in the context of the state of the art. The problematic behavior of the state-of-the-art architecture is well illustrated in Figures 1 and 2. In general, the illustrations and explanations allow a non-expert to understand the specificity of hierarchical classification, the state-of-the-art choices, and the resulting problems. I also appreciated the care taken in explaining the metrics to understand what they represent and how they complement each other.

The solution proposed by the authors makes sense, as they have found that CAST's architecture is very well suited to solve this problem. The explanations and schematics of the architecture are clear, I just had to go to the CAST article to understand how super-pixel pooling is done (which is not critical to understanding the article). The two losses also make sense, as the authors realized that the problem of the state of the art comes from an independent resolution at each level of the hierarchy, and added a loss that forces a (more) globally correct classification. The experimental part (Figure 4, TICE metric) shows that their model makes fewer structural errors (compared to the tree structure), even if it is smaller.

Table 2 and the different ablations clearly show the impact of the improvements made. The last part, which shows that hierarchical learning can improve segmentation, is an interesting addition, which may lead us to think of an opening towards hierarchical segmentation (for which super-pixel-based embedding seems to be well suited).

On a more personal note, at a time when the tendency is to build gigantic, high-consumption networks, I find it appreciable to see methods demonstrating that designing a solution specifically for a problem can increase its efficiency.

**Weaknesses:**

In part 4.4, I'm not sure how to interpret the result. We can assume that the result is a failure if the superpixel segmentation doesn't make sense. But to me, this just shows that superpixel segmentation is not necessarily adapted to a semantic problem, because it's done at the color level, and semantic in images is not necessarily associated with color.

In Figures 1 and 4, I find that the space between class names (like "Chimpanzeeferret" in Figure 1) and their position/alignment in the tree can be difficult to read. In two-word classes, you might want to break a line, because at first sight the second part of the name seems to be another class link to another node.

**Questions:**

The following paragraphs are meant to be an open reflection. I'm not an expert in hierarchical classification, so I may have missed some key parts of the problem, modeling, and overall reflection.

I think fine-to-coarse works well here because it fits the network architecture, not because it's a better choice overall. Indeed, if the hierarchical structure of the labels is a tree, one could simply put all the effort into classifying the fine level and infer the higher levels by going up the tree, since each class has only one parent. Of course, this would mean that a single error on the fine level would invalidate all other classifications.

Overall, I think the coarse-to-fine direction makes more sense, since it's all about descending a tree and thus reducing the search field of the lower layers. I just don't think this architecture is suited for that. However, it may be possible to design a sequential architecture in coarse-to-fine logic, for example a U-net. We could imagine a U-net or an encoder-decoder using the CAST structure with a symmetric decoder. It would then be necessary to find a way to divide the super-pixels. The classification will be perform sequentially by going up the decoder

Whether it's one way or the other (but rather the other), I think that the sequential classifications should directly influence each other (like in a Markov chain or a transition table), and not just as different normalizations of the classification probabilities.

---

> ### Author Response · Authors · 2024-11-22
>
> Dear reviewer YARY,
> Thank you for your valuable feedback and detailed  comments. We greatly appreciate your thorough understanding of our work and your thoughtful recognition of its strengths. Your remarks on the clarity of our problem setup, method design, and experimental results are deeply encouraging. We address your concerns and questions in the response below.
>
>
> ---
> **1. Interpretation of Part 4.4**
> >In part 4.4, I'm not sure how to interpret the result. .... But to me, this just shows that superpixel segmentation is not necessarily adapted to a semantic problem, because it's done at the color level, and semantic in images is not necessarily associated with color.
>
> As you mentioned, the purpose of the superpixel segmentation is not to directly associate colored segments with specific semantics. Instead, we use it to evaluate whether the model effectively groups the desired object, which indirectly reflects the quality of its predictions.
>
> We replaced the example images with ones that are easier to understand. For example, in the **updated PDF’s Figure 5**, the first row shows two images where the model is tasked with recognizing a shoe. In the correct prediction case, the shoe and the sock are well-grouped (in green), showing coherent segmentation. In the incorrect prediction case, despite being a similar image, the model fails to recognize these parts, resulting in highly fractured segments.
>
> While superpixel segmentation operates at the color level and may not fully capture semantic meaning, these examples **help illustrate how the model’s focus aligns with its predictions**. This provides indirect interpretability, allowing us to understand the reasoning behind correct and incorrect predictions.
>
> We hope this explanation clarifies the intent and interpretation of these results. Please let us know if further clarification is needed.
>
>
> ---
> **2. Modification of Figure 1 and 4**
> > In Figures 1 and 4, I find that the space between class names (like "Chimpanzeeferret" in Figure 1) and their position/alignment in the tree can be difficult to read. ...
>
> Thank you for the careful review. We have revised it accordingly.
>
> ----
> **3. Reflection on Fine-to-coarse design**
>
> Thank you for the insightful reflections. Your points on the coarse-to-fine direction are valid, and we appreciate the opportunity to further clarify the rationale behind our fine-to-coarse approach.
>
> Firstly, as the reviewer mentioned, if the hierarchical structure of labels is a tree, it could make sense to classify the fine level first and infer the higher levels by traversing up the tree (i.e., a flat classification approach). However, as you correctly noted, a single error at the fine level would invalidate all higher-level predictions. This highlights the need for a model capable of classifying all labels within the hierarchy.
>
> As we included the experiments on Coarse-to-Fine and Fine-to-Coarse architectures in Table 3, we carefully considered how to design the architecture effectively. Since taxonomies often follow a tree structure, with broader classes leading to finer categories like genus and species, the coarse-to-fine approach naturally aligns with human perception and has been widely adopted in prior works.
>
> However, taxonomy is a human construct, and it made us question whether machine learning models should necessarily process information in a coarse-to-fine manner. When we reflect on how we learn abstract/coarser concepts, we often find that abstract categories can be learned  more easily by learning specific concepts. For instance, seeing Siberian Huskies, Chihuahuas, Malteses and so on could lead to the broader category "dog" based on shared features like being "cute, four-legged animals."
>
> This perspective, along with the structural design we adopt from CAST, inspired us to explore a Fine-to-Coarse learning strategy, where fine features are aggregated to form higher-level features (segments). Surprisingly, this approach achieved strong performance in our experiments, suggesting its potential for hierarchical classification.
>
> We believe the best model for hierarchical classification is still underexplored, and as you suggested, there is much potential to investigate more direct and diverse approaches. Thank you again for the thoughtful discussion—it has inspired us to think further on these possibilities.

---

> > ### Comment · Reviewer_YARY · 2024-11-26
> >
> > Dear authors, thank you for your response, my few concerns have been addressed.

---

> > > ### Author Response · Authors · 2024-11-26
> > >
> > > We’re pleased to hear that some of your concerns have been addressed, and we sincerely appreciate your recognition of our contributions.

---

### Official Review · Reviewer_khL4 · 2024-10-30

**Soundness:** 2
**Presentation:** 2
**Contribution:** 2
**Rating:** 5
**Confidence:** 5

**Summary:**

This work proposes H-CAST, a model built upon CAST for hierarchical image classification, by addressing both visual consistency and semantic consistency across predictions at different hierarchical levels. To achieve this, hierarchical supervision at different network layers and a tree loss are introduced. Experiments on three datasets verify the proposed method can achieve better performance than the baseline method.

**Strengths:**

1. The motivation for visual consistency is sound.
2. This work introduces new metrics for hierarchical classification, which can measure the coherence of hierarchical predictions.
3. The proposed method achieves SOTA on all datasets.

**Weaknesses:**

1. The key point of this paper, which is making classifiers at different levels attend to consistent visual cues, lacks support from quantitative results or theory. For example, if wrongly classified cases are all associated with incorrect CAMs? What is the transfer rate after adopting the proposed model? Qualitative comparisons are subjective and lack statistical significance. In addition, Grad-CAM is an approximation and does not truly explain how the network operates.
2. This work relies heavily on CAST. Though the authors propose the concept of visual consistency, its implementation is directly borrowed from CAST. Consequently, the primary contributions of this work are merely the hierarchical supervision loss and tree KL loss, neither of which can ensure visual consistency. Additionally, the fine-to-coarse training strategy is also derived from CAST. The tree-path KL loss is trivial.
3. The tree KL loss defines the ground truth distribution according to the number of hierarchical levels (i.e., 1/L). It would be beneficial to address whether the proposed solution can effectively manage larger hierarchies, particularly those with depths of up to 10 or 20.
4. While the clustering module would potentially guide the model to attend to spatially-coherent regions, it is unclear why the model would “ensure that each hierarchical classifier focuses on the same corresponding regions”. In fact, the model still has the opportunity to find shortcuts (attending to different regions in different levels as Fig. 2) and meanwhile deliver correct classification results. In addition, I expect visualizations of Grad-CAM results in different hierarchical levels from the proposed model.
5. The experiments are somewhat questionable. Firstly, the latest hierarchical competitor, HRN, published in CVPR'22, is relatively outdated. Secondly, the experimental results of HRN differ from those reported in the original publication. Thirdly, there are several competitors [a-b] (might be more, not carefully checked), released before two months prior to the DDL, that outperform this work.
6. Results in Fig. 5 do not totally make sense to me. Examples in a) and b) are not equally recognizable, i.e., all examples in b) are much harder to distinguish/group than those in a). As a result, it is hard to confirm whether poor clustering of pixels in b) is the cause or the effect of incorrect predictions. One way for improvement is to examine for similar hard-level images in a) whether correct predictions are achieved along with better clustering results.
7. According to [c], existing work has explored various loss functions for hierarchical classification. It would be useful to compare the effectiveness of tree KL loss against these alternatives. Of course, the analysis should also be provided.
8. The comparison to hierarchical approaches is unfair. While FGN and HRN use ResNet-50 as the backbone， this work adopts ViT-S.
9. More top-leading hierarchical classification work should be included in the comparison.
10. The datasets used for evaluation are small. larger datasets with complex hierarchy (ie.g., ImageNet-1K, iNaturalist) should be evaluated to better assess effectiveness.
11. How about the training and inference speed of the proposed method? Given the incorporation of superpixels and segmentation in addition to classification, it is necessary to provide a comparison of resource costs, including both time and memory usage.
12. The literature review is far from complete. In addition to [a, b, c], numerous efforts in hierarchical scene parsing is totally missing; see  related work section in [d, e]. As a top-conference paper, a comprehensive literature review is a basic requirement. I believe the reference part should be greatly extended.
13. In fact, the hierarchical loss function in [d] is superior to the proposed Tree-PATH KL loss, in that it guarantees hierarchy-aware coherent predictions while the proposed loss cannot. A strict quantitative comparison of the two loss functions should be provided.
14. Minor issues include: a) the usage of the terms "interpretability" (L447) and "explainability." b) citation format. c) vector images. d) missing period after Eq. 2. e) the presentation of Eq. 3.
15. A Conclusion section should be added to properly conclude the work and offer insights of downside of impact of the work.
[a] HLS-FGVC: Hierarchical Label Semantics Enhanced Fine-Grained Visual Classification.
[b] Hierarchical multi-granularity classification based on bidirectional knowledge transfer.
[c] Hierarchical classification at multiple operating points. NeurIPS 2022
[d] Deep Hierarchical Semantic Segmentation. CVPR 2022
[e] LogicSeg: Parsing Visual Semantics with Neural Logic Learning and Reasoning, ICCV 2023

**Questions:**

Please see the above.

---

> ### Author Response · Authors · 2024-11-22
> **Official Comment by Authors (1)**
>
> Dear reviewer khL4,
> Thank you for your valuable feedback and comments. We appreciate your recognition of the sound motivation for visual consistency, the new metrics for hierarchical classification, and the state-of-the-art performance of our method. Your diverse and constructive comments have been instrumental in improving our work. We have also updated the PDF to reflect these improvements, and we kindly invite you to review the revised version. Below, we address your concerns and questions.
>
> ---
> **1-1. Quantitative Support for Consistent Visual Grounding**
> >1. The key point of this paper, which is making classifiers at different levels attend to consistent visual cues, lacks support from quantitative results or theory.
>
> To provide quantitative support for our observation in Figure 2, we analyzed Grad-CAM heatmaps of coarse and fine-grained classifiers.
> Specifically, we compute  two metrics: the overlap score and the correlation score. The **overlap score** quantifies the degree to which the regions activated by the two classifiers coincide. The **correlation score** measures the linear relationship between the activation values of the overlapping regions in the two heatmaps.
> Higher overlap and correlation scores indicate stronger agreement between the regions attended to by the two classifiers. Conversely, lower scores highlight a lack of alignment in their focus.
>
> In the Table below, results from the FGN model on the Entity-30 dataset show that, interestingly, when both classifiers made correct predictions, overlap and correlation scores were significantly higher. Conversely, incorrect predictions corresponded to notably lower scores. These findings support our motivation that aligning the focus of classifiers can enhance both accuracy and consistency. We have updated the detailed explanation and results in Appendix A.
>
>
> | Overlap Score |  | Fine | pred. |
> |---|---|---|---|
> |  |  | True | False |
> | **Coarse** | True | **0.51 &pm; 0.20** | 0.25 &pm; 0.13 |
> | **Pred.** | False | 0.36 &pm; 0.18 | 0.37 &pm; 0.19 |
>
> | Correlation |  | Fine |  Pred. |
> |---|---|---|---|
> |  |  | True | False |
> | **Coarse** | True | **0.70 &pm; 0.26** | -0.02 &pm; 0.40 |
> | **Pred.** | False | 0.30 &pm; 0.42 | 0.35 &pm; 0.41 |
>
>
> **1-2. Utility and Limitations of Grad-CAM and Transfer Rate**
> >What is the transfer rate after adopting the proposed model? In addition, Grad-CAM is an approximation and does not truly explain how the network operates.
>
> While Grad-CAM is an approximation, it is a widely used tool for observing class activation and provides valuable insights into model behavior. Our analysis demonstrates meaningful patterns that support the effectiveness of our proposed method. However, we acknowledge its limitations and will explore additional evaluation methods in future work.
>
>
> If the reviewer refers to the improvement in consistency and accuracy after adopting our method, our experimental results already demonstrate superior and consistent performance across benchmark datasets, indirectly supporting the effectiveness of consistent visual grounding. If clarification is needed, we are happy to provide further details.
>
> ---
>
> **2. Novel Contribution over CAST**
>
> We thank the reviewer for the opportunity to clarify our contributions to hierarchical classification and the novel role of CAST in our work.
>
> (1) **Key Insight**:
> Our observation revealed that classification at different granularities involves fundamentally distinct tasks requiring attention to different but consistent regions within an image. We found that inconsistencies arise because each classifier tends to independently attend to different regions without connection. This observation led us to propose consistent visual grounding as a novel solution to connect hierarchical classifiers across levels.
>
> (2) **Leveraging Semantic Segments for Hierarchical Classification**:
>    While we adopted CAST as part of our architecture, it is important to emphasize that CAST originates from a different task, weakly-supervised semantic segmentation. In CAST, “hierarchy” refers to “part-to-whole” visual grouping (e.g., eyes, nose, arms), while our work addresses a “taxonomy hierarchy” (e.g., bird - Green Hermit). It was NOT evident that the concept of “part-to-whole” segments would align well with a taxonomy hierarchy; this connection is a novel discovery introduced through our work.
>
> In addition, based on our observation in (1), we newly propose leveraging segments at different granularities to enhance multi-granularity classification. To the best of our knowledge, the use of segments has NOT been applied to hierarchical classification tasks.
>
> Thus, this adaptation is neither trivial nor an obvious solution; it stems from our novel observation and bridges two distinct fields to tackle challenges unique to hierarchical classification.
>
> We hope this summary highlights the novelty and importance of our work.

---

> ### Author Response · Authors · 2024-11-22
> **Official Comment by Authors (2)**
>
> **3. Larger hierarchies with Tree-path KL loss**
> > It would be beneficial to address whether the proposed solution can effectively manage larger hierarchies, particularly those with depths of up to 10 or 20.
>
>
> Thank you for the insightful question. Managing larger hierarchies with depths of 10 or 20 is indeed an important point to consider. While our current approach focuses on 2-3 levels, following prior hierarchical multi-granularity classification works [1, 2, 3], scaling to deeper hierarchies poses new challenges. Specifically, applying our current KL loss directly to such deep levels would likely encounter difficulties in maintaining effectiveness.
>
> We acknowledge the need for scalability in deeper and more imbalanced trees and view this as a promising direction for future research. Exploring adjustments to the loss function and other architectural adaptations to handle larger hierarchies is an exciting area we plan to investigate further.
>
> [1] Your “Flamingo” is My “Bird”: Fine-Grained, or Not, 2021
> [2] Consistency-aware feature learning for hierarchical fine-grained visual classification, 2023
> [3]  Hierarchical multi-granularity classification based on bidirectional knowledge transfer, 2024
>
> ---
>
> **4. Visualization of Attention maps**
> > ... it is unclear why the model would “ensure that each hierarchical classifier focuses on the same corresponding regions” ... I expect visualizations of Grad-CAM results in different hierarchical levels from the proposed model.
>
> We acknowledge that the model may sometimes use shortcuts by attending to different regions at different levels and still deliver correct predictions. Our method is designed to guide classifiers toward consistent visual grounding, but it does not directly enforce this behavior. To better reflect this, we will tone down our wording from “ensure” to “guide” in the manuscript.
>
> Instead of Grad-CAM, we visualized attention maps from the transformer, as they provide a more direct representation of what the model attends to. The visualizations reveal that from lower to upper blocks, the model increasingly attends to similar regions. In the lower blocks, attention is more detailed and localized (e.g., snake’s head, parts of its body), while in the upper blocks, attention expands to include broader regions encompassing the areas highlighted by the lower blocks. These patterns align with our intended design for visual grounding in hierarchical classification.
>
> We believe these visualizations validate our claim and have included them in the Appendix D.2.
>
> ---
>
> **5. 8. 9. Experimental Comparisons and Baseline Choices**
> >5. The experiments are somewhat questionable... HRN, published in CVPR'22, is relatively outdated... several competitors [a-b] ...
>
> >9. More top-leading hierarchical classification work should be included in the comparison.
>
> Thank you for introducing the new research [a, b]. We have added it to the related work section. While we aimed to compare our method with the most recent high-performing studies, regrettably, none of the latest works [a, b, c], including those you mentioned, had publicly available code. Also, this limitation is partly due to the focus of most studies on flat-level classification, resulting in few baselines for hierarchical multi-granularity classification. We hope our work inspires further research in this important area.
>
>
> >8.  While FGN and HRN use ResNet-50 as the backbone, this work adopts ViT-S.
>
> To account for differences due to backbone choices, we explicitly indicated the backbone architecture in the tables. As there were no existing hierarchical multi-granularity classification works using a ViT backbone, we introduced Hier-ViT to provide a ViT-based comparison. Additionally, as strong baselines, we trained flat-level classifiers and applied HiE [d] to improve fine-grained classification using coarse classifiers.
>
> > 5. ... the experimental results of HRN differ from those reported in the original publication.
>
> Upon review, we noticed that the results for the CUB and Aircraft datasets were reported using a batch size of 64 instead of 8. We have corrected this to the results with a batch size of 8. Thank you for your careful attention to detail. Also, slight differences occur because the original paper trained for 200 epochs, whereas we standardized all experiments to 100 epochs for fairness. Beyond this, we followed their codebase and experimental settings.
>
>
> We hope this clarifies our experimental comparisons and addresses the reviewer’s concerns.
>
>
> [a] HLS-FGVC: Hierarchical Label Semantics Enhanced Fine-Grained Visual Classification, 2024
> [b] Hierarchical multi-granularity classification based on bidirectional knowledge transfer, 2024
> [c] Consistency-aware feature learning for hierarchical fine-grained visual classification, 2023
> [d] Test-Time Amendment with a Coarse Classifier for Fine-Grained Classification, 2023

---

> ### Author Response · Authors · 2024-11-22
> **Official Comment by Authors (3)**
>
> **6. Comparison between Correct and Incorrect Predictions**
>
> > 6. Results in Fig. 5 do not totally make sense to me. Examples in a) and b) are not equally recognizable, ... One way for improvement is to examine for similar hard-level images ....
>
>  Initially, the images were selected randomly; however, in response to the reviewer’s suggestion, we have updated the PDF to include examples of images with comparable difficulty levels. These examples consistently demonstrate that correct predictions exhibit better clustering, while incorrect predictions often show fractured or misaligned groupings.
>
> For example, in the updated PDF’s Figure 5, the first row shows two images where the model is tasked with recognizing a shoe. In the correct prediction case, the shoe and the sock are well-grouped (in green), showing coherent segmentation. In the incorrect prediction case, despite being a similar image, the model fails to recognize these parts, resulting in highly fractured segments.
>
> While Figure 5 does not explicitly determine whether poor clustering is the cause or effect of incorrect predictions, it highlights a clear relationship that provides valuable insights into the model’s reasoning process. We have added additional examples in Appendix Figure 8.
>
> ---
> **7, 13. Justification of Tree-path KL divergence loss and comparison with other losses**
> >7. .. It would be useful to compare the effectiveness of tree KL loss against these alternatives...
>
> >13. In fact, the hierarchical loss function in [d] is superior to the proposed Tree-PATH KL loss ..
>
> To evaluate the effectiveness of our Tree-path KL Divergence loss, we compared it with two alternatives: Binary Cross Entropy (BCE) loss in [c] and Flat Consistency loss. BCE directly replaces the KL divergence component, while Flat Consistency loss, inspired by a bottom-up approach, infers coarse predictions from fine-grained ones and uses BCE to match them with the ground truth.
>
> As shown in the table below, Tree-path KL Divergence loss outperforms both alternatives, achieving the highest FPA on the Living-17 dataset and demonstrating superior accuracy and semantic consistency. Similar trends are observed on the Aircraft  dataset.
>
> We have included these results and explanations in the updated manuscript  Table 5 and 10.
>
> **<Living-17 dataset>**
>
> | Semantic consistency loss | FPA | Coarse | Fine | wAP |
> |---|---|---|---|---|
> | Flat consistency loss | 82.82 | 88.88 | 83.53 | 85.31 |
> | BCE loss | 83.65 | 89.76 | 84.00 | 85.92 |
> | KL Divergence loss | **85.12** | **90.82** | **85.24** | **87.10** |
>
>
> **<Aircraft dataset>**
>
> | Semantic consistency loss | FPA | maker | family | model | wAP |
> |---|---|---|---|---|---|
> | Flat consistency loss | 82.87 | 94.63 | 90.94 | 84.97 | 88.51 |
> | BCE loss | 82.18 | 94.21 | 90.13 | 84.88 | 88.11 |
> | KL Divergence loss | **83.72** | **94.96** | **91.39** | **85.33** | **88.90** |
>
>
>
> Regarding the loss in [d], it appears to focus on pixel-level hierarchical segmentation tasks, which are not directly applicable to our instance-level classification setting. We hope this clarifies our approach and the scope of the comparisons.
>
> ---
>
> **10. Experiments on larger datasets**
>
> > 10. The datasets used for evaluation are small. larger datasets with complex hierarchy (ie.g., ImageNet-1K, iNaturalist) should be evaluated to better assess effectiveness.
>
> We would like to clarify our dataset choices and experimental setup.
>
> First, CUB, Stanford Cars, and Aircraft are among the most widely used benchmark datasets in prior hierarchical classification studies [1, 2, 3, 4]. We selected these datasets to ensure a fair comparison with existing methods. To further evaluate the effectiveness of our approach on a more diverse and challenging dataset, we conducted experiments on BREEDS, a subset of ImageNet, which includes a broader range of classes beyond a single type (e.g., birds or aircraft).
>
> As for ImageNet-1K, its highly imbalanced hierarchy poses significant challenges for applying our approach. Consequently, most hierarchical classification studies on this dataset have focused on flat-level classification rather than multi-granularity approaches.
>
> To further validate our method, we are currently conducting additional experiments on the iNaturalist dataset, which provides a larger and more complex test bed. We will update the results as soon as they become available. However, we kindly request your understanding, as our experiments are conducted on a single GPU system (Nvidia A40), which leads to longer training times for larger datasets.
>
>
> [1] Your “Flamingo” is My “Bird”: Fine-Grained, or Not, 2021
> [2] Consistency-aware feature learning for hierarchical fine-grained visual classification, 2023
> [3]  Hierarchical multi-granularity classification based on bidirectional knowledge transfer, 2024
> [4] HLS-FGVC: Hierarchical Label Semantics Enhanced Fine-Grained Visual Classification, 2024

---

> ### Author Response · Authors · 2024-11-22
> **Official Comment by Authors (4)**
>
> **11. Resource costs**
> > 11, How about the training and inference speed of the proposed method?  ... it is necessary to provide a comparison of resource costs, including both time and memory usage.
>
> As suggested by the reviewer, we have included training/inference time and memory usage to the table below. The time is measured per iteration with a batch size of 64 on an NVIDIA A40 GPU. In terms of memory usage, graph pooling reduces the size of the patches, leading to  lower memory consumption compared to using patches of the same size without pooling. However, the additional computation required for superpixel generation and graph pooling increases the time compared to a standard ViT.
>
> Our method achieves higher performance (>10%) than a standard ViT but comes with the trade-off of increased computation time. We have included this limitation in the Discussion section.
>
> Additionally, the current implementations of graph pooling and superpixel generation are not fully optimized, and we expect future improvements with more efficient algorithms to address this limitation.
>
> |  | Hier-ViT | H-CAST |
> |---|---|---|
> | Memory Usage (GPU) | 3.7GB | 3.1GB |
> | Training/Inference time | 0.7s | 1.2s |
>
> ----
>
>
> **12. More thorough literature review**
> >12. The literature review is far from complete. In addition to [a, b, c], numerous efforts in hierarchical scene parsing is totally missing; see related work section in [d, e].
>
> We have revised the related work section to clearly distinguish our work from existing studies and to provide a more comprehensive overview of the field. Specifically, we have incorporated discussions on hierarchical semantic segmentation, including the works mentioned [d, e], as well as additional references to ensure coverage of hierarchical scene parsing efforts.
>
> Due to space limitations, we have kept the main text concise while adding more detailed discussions in the Appendix. We believe this revision addresses the reviewer's concern and provides a clearer context for our contributions.
>
> ----
>
> **14. Minor issues**
> >14. Minor issues include: a) the usage of the terms "interpretability" (L447) and "explainability." b) citation format. c) vector images. d) missing period after Eq. 2. e) the presentation of Eq. 3.
>
> We have addressed all other points, but could you clarify what is meant by "a) the usage of the terms 'interpretability' and 'explainability' (L447)”? This clarification will help us ensure an appropriate response.
>
> ----
> **15. Conclusion Section**
> > A Conclusion section should be added to properly conclude the work and offer insights of downside of impact of the work.
>
>  We have added a Conclusion section to summarize the work and discuss the limitations.
>
> ---------
> We hope this clarifies the reviewer’s concerns. If there are any further concerns or clarifications needed, we would be happy to discuss them further.

---

> ### Author Response · Authors · 2024-11-25
> **Update on larger dataset**
>
> **Experiments on larger dataset, iNat21-Mini**
>
>
> Thank you for waiting. We'd like to share the results of our experiments on the large-scale dataset (iNaturalist 2021-mini). iNat21-mini [1] contains a total of *10,000 classes* and *500,000 training samples* and *100,000 test samples*, structured within an 8-level hierarchy. For our experiments, we strategically focused on a 3-level hierarchy consisting of *name, family, and genus*. We made this choice because we believe that models with a meaningful level of granularity are more practical for real-world applications.
>
> To elaborate, the number of classes at each level is as follows:
> Kingdom: 3, Supercategory: 11, Phylum: 13, Class: 51
> Order: 273, Family: 1,103, Genus: 4,884, Name: 10,000
>
> We deliberately excluded extremely coarse-grained levels like *kingdom* (3 classes), as such distinctions offer minimal practical value for classification tasks. Likewise, overly fine-grained levels such as *genus* (4,884 classes), where many species are represented by only one or two samples, fail to offer meaningful differentiation from direct *name*-level classification.  Thus, we selected **order** (**273** classes), **family** (**1,103** classes), and **name** (**10,000** classes) for our 3-level hierarchy. This choice ensures that **each higher-level class meaningfully represents a diverse yet relevant subset of lower-level classes**, enabling both meaningful classification and the evaluation of consistent predictions.
>
> The results are presented in the table below. Compared to Hier-ViT, which uses the same ViT-small backbone, our method demonstrates the **fine-grained accuracy improvement of over 7.29%** and a **8.3% improvement in the FPA metric**, representing a significant performance gain.
>
> We hope these results address concerns about large-scale data.
> Also, while our experiments focused on a meaningful 3-level hierarchy, as previously addressed in our response to Comment 3, we believe that designing a model capable of efficiently scaling to deeper hierarchies represents an important and promising direction for future work.
> We will include this experimental result in the revised manuscript.
>
> **< iNat21-Mini (273 - 1,103 - 10,000) >**
> |  | FPA | Order | Family | Name | wAP | TICE |
> |---|:---:|:---:|:---:|:---:|:---:|:---:|
> | Hier-ViT | 56.73 | 87.54 | 79.79 | 62.81 | 65.05 | 24.34 |
> | Ours | **65.03** | **89.84** | **84.12** | **70.09** | **71.92** | **15.92** |
>
> [1]  Benchmarking representation learning for natural world image collections. 2021

---

> > ### Comment · Reviewer_khL4 · 2024-11-26
> > **Thanks for the response, but I still feel the novelty is not enough**
> >
> > Despite some clarifications, I still believe that this work does not meet the standards of ICLR.
> >
> > Regarding the novelty, I share similar a view with Reviewers uxN6 and ZNGF. I feel the novelty is limited and the discussions are not insightful.
> >
> > Regarding the experiments, the authors exclude many recent works. Even the code is not released, the authors should reimplement the algorithm. In addition, the authors arbitrarily change the number of training epochs, and report the results with different batch size, making me feel the results unconvinced.
> >
> > Moreover, the authors state that "Regarding the loss in [d], it appears to focus on pixel-level hierarchical segmentation tasks, which are not directly applicable to our instance-level classification setting." Although [d] addresses hierarchical segmentation, it is very clear that the loss used in [d] can be applied for image-level classification. I am very sure about this. This also shows the limited knowledge of the authors about this field.
> >
> > Finally, the authors state "In CAST, “hierarchy” refers to “part-to-whole” visual grouping (e.g., eyes, nose, arms), while our work addresses a “taxonomy hierarchy” (e.g., bird - Green Hermit). " I do not think this is a big different.  "part-to-whole" and "taxonomy hierarchy" both can be seen as types of concept taxonomy.
> >
> > Given these fundamental issues, I will maintain my score.

---

> ### Author Response · Authors · 2024-11-26
>
> We appreciate the reviewer’s feedback and would like to clarify and address the following concerns.
>
> **<Experiments>**
>
> While some recent works are not included due to the lack of publicly available code or model checkpoints, it is important to note that all these works are also based on **ResNet backbones**. Given this, we believed it was more appropriate to include a ViT-based baseline rather than another ResNet-based one. Furthermore, since no prior works in multi-granularity classification used a ViT backbone, we developed Hier-ViT to fill this gap.
>
> Regarding **training epochs**, we standardized all experiments to 100 epochs to ensure fair comparisons. FGN originally used 100 epochs, so we maintained this setting, and we reduced HRN from 200 epochs to 100 for consistency. Standardizing training epochs is a widely accepted practice for fair evaluation. Additionally, our results for FGN show improved performance compared to the results reported in their paper, and for HRN, the performance differences are minimal, ranging from 0.09 to 0.47.
>
> For HRN, we included experiments with **a larger batch size**, along with the original batch size of 8, because the method exhibited significant sensitivity to batch size, a notable observation that highlights its behavior compared to other methods.
>
> Based on these considerations, we respectfully disagree with the claim that our experimental setup is unconvincing. We believe our decisions ensure a fair and meaningful comparison.
>
> ---
> **<Loss in Hierarchical Segmentation [d]>**
>
> We appreciate the opportunity to clarify our statement regarding the loss in [d]. While it is indeed possible to adapt the loss in [d] for instance-level classification by treating instances as analogous to pixels, **our primary intention was to emphasize the fundamental differences in focus between the two tasks**.
>
> In [d], during inference, each pixel \(i\) is associated with the top-scoring root-to-leaf path in the class hierarchy \(T\). These **root-to-leaf paths are predefined, and the task focuses on selecting the best path**. As a result, the loss in [d] is designed to emphasize the weakest predictions along the path to improve overall accuracy. **In contrast, our objective is to predict *each node* along the root-to-leaf path *individually***, ensuring that all predictions are **both accurate and consistent across the hierarchy**. The key challenge in our task lies in addressing potential **inconsistencies between hierarchy levels**, and we propose a method specifically designed to resolve this issue.
>
> Specifically, **the loss in [d]** prioritizes the most violated hierarchical constraint for each score vector through the "**min**" operation in Equation (6).
> That means, if the loss (6) in [d] is applied, the model may prioritize resolving constraints for the fine-grained taxonomy (e.g., "Green Hermit") at the expense of optimizing the coarse taxonomy (e.g., "bird").
> In contrast, our approach is **designed to simultaneously predict multiple taxonomies across the hierarchy** (e.g., "bird" and "Green Hermit").
> Thus, in our approach, we **model all classes at each level as distributions** and adopt a KL divergence loss to encourage balanced learning across all taxonomy levels. This ensures a holistic approach that aligns with the multi-granularity objectives of our task.
>
> Nonetheless, we acknowledge the potential utility of explicitly enforcing hierarchical constraints through the loss in [d] and will conduct experiments to evaluate its applicability in the instance-level setting.
>
> ----

---

> ### Author Response · Authors · 2024-11-26
>
> **<Concept hierarchy>**
>
> We appreciate the reviewer’s comment and would like to clarify the distinction between **part-to-whole hierarchies** and **taxonomy hierarchies**, as well as how our approach bridges these concepts. While both are conceptual hierarchies, their foundations and applications are fundamentally different.
>
> A **part-to-whole hierarchy** represents a **spatial compositional hierarchy**, where smaller parts (e.g., eyes, nose, arms) combine to form a larger whole (e.g., a face or body). This type of hierarchy is grounded in **visual composition and spatial relationships**. In contrast, a **taxonomy hierarchy** (e.g., "bird" → "Green Hermit") is a **semantic hierarchy**, structured by coarse-to-fine class relationships that are defined by **meaning and semantics**, not by spatial composition.
>
> Our work bridges these two ideas by **incorporating visual grounding concepts**—which are typically applied in spatial compositional hierarchies (segmentation tasks)—to address **challenges in semantic taxonomy hierarchies** (hierarchical classification task).
>
> On that distinction, **exiting works only enforce the consistency along the semantic hierarchy, whereas ours is the only one that ground the consistency of semantic hierarchy on the visual spatial parsing consistency**.  As a consequence, our work outperforms Hier-ViT  (a ViT-based model that enforces only semantic consistency)  by more than 10%, a significant margin, and SOTA (HRN) by over 4.25-6.36% on BREEDS, subset of ImageNet with more diverse categories.
>
> On the benchmark of hierarchical classification, we deliver the **significant gain** with the **first (unsupervised) visually grounded classification model**.  Our experimental validation is solid and our model stands out in novelty as a single such paper on the topic of hierarchical classification.
>
> We urge the reviewers not to let this seeming conceptual resemblance overlook our significant contributions on both accounts (practical results and vision insight).

---

> > ### Comment · Reviewer_khL4 · 2024-11-28
> >
> > Thanks for the reply. After viewing all the reviewers' comments and the responses from the authors, I feel this work is clearly far from the bar of ICLR. I vote for reject.

---

### Official Review · Reviewer_d3Ey · 2024-11-04

**Soundness:** 2
**Presentation:** 2
**Contribution:** 2
**Rating:** 6
**Confidence:** 4

**Summary:**

The paper proposes H-CAST architecture for hierarchical classification tasks. The architecture builds on top of prior CAST work: superpixels are fed into a ViT network, where periodic graph pooling operation aggregates the tokens of high similarity. This produces a fine-to-course hierarchy of features. Linear layers at each level of the hierarchy are used as classification heads. The paper also presents tree-path KL loss, where the entire path in the hierarchical class tree is matched. The method shows strong performance over baselines in several benchmark datasets.

**Strengths:**

(1) The model shows better results than prior works.

(2) The examples given in the introduction help to explain and illustrate the reasoning behind the hierarchical focus.

(3) The ablations confirm that the added additional loss contributes to the performance.

**Weaknesses:**

(1) In L61, the difference in available labelling is presented as one of the motivations for hierarchical classification. However, the presented method assumes that all levels of the hierarchy are available. Can the technique work if the finest levels of supervision are not available?

(2) Similarly, given the availability of the finest-level label, the other course levels in the tree are implied, so perhaps a more appropriate flat baseline would be a ViT that only predicts finest-level classes (and thus parent nodes by simple aggregation). It would also provide a more appropriate comparison in terms of architecture.

(3) Given the relatively "small" sizes of the datasets (Tab 1.) by modern standards and some occasional closeness to the flat baselines in the scores (Tab 2.) Has there been any significant variability observed in the results? Would it be possible to include some measure of variance for some key results?

**Questions:**

Please see questions listed alongside weakness.

---

> ### Author Response · Authors · 2024-11-22
>
> Dear reviewer d3Ey,
>
> Thank you for your valuable feedback and comments. We appreciate your recognition of the motivation behind the hierarchical focus, the strong results compared to prior works, and the insights provided by the ablation studies. We address your concerns and questions in the response below.
>
> ---
> To address questions (1) and (2), we would like to clarify the two main directions in hierarchical classification:
> 1. **Flat Classification**: This approach assumes a known taxonomy and focuses on fine-grained (flat-level) classification. Coarse labels are used during training to improve fine-grained predictions. At inference, higher-level taxonomy is derived in a **bottom-up manner** from fine-grained predictions. The **output is a single label**, and most existing works adopt this approach. While effective for detailed and clear images (e.g., close-up shots of birds), it can struggle with less distinguishable objects, as errors at the fine-grained level often lead to incorrect predictions at higher levels.
>
> 2. **Global (Multi-granularity) Classification**: This approach, which includes our work, predicts the **entire taxonomy**, addressing the limitations of fine-grained classification. By providing higher-level classifications, it offers more flexibility and robustness in real-world scenarios with ambiguous or partially visible objects.
>
> ---
> **(1) Supervision Assumption**
> > (1) In L61, the difference in available labelling is presented as one of the motivations for hierarchical classification. However, the presented method assumes that all levels of the hierarchy are available. Can the technique work if the finest levels of supervision are not available?
>
> In L61-62, we aimed to emphasize the importance of full-taxonomy classification under realistic scenarios. Specifically, L61 illustrates cases where coarse labels, like "bird," may suffice for some users, but experts require finer distinctions. We have revised this section to make it clearer (L33-L39).
> For this work, we assumed all labels are available during training (fully supervised). Your inquiry about the absence of fine-level labels during training is valid and highlights an important area for future exploration. Semi-supervised scenarios, where some labels are unavailable, represent another interesting and challenging problem that could extend this work.
>
> ---
> **(2) Flat Baseline Comparison**
> > (2) Similarly, given the availability of the finest-level label, the other course levels in the tree are implied, so perhaps a more appropriate flat baseline would be a ViT that only predicts finest-level classes (and thus parent nodes by simple aggregation). It would also provide a more appropriate comparison in terms of architecture.
>
> Your observation about the baseline aligns with the bottom-up inference in fine-grained classification. We have already included flat-level baselines trained at each level, as shown in Tables 2, 11, 12 in updated pdf (Flat-ViT/Flat-CAST). For this discussion, we can focus on the fine-level results, assuming that fine-level predictions are aggregated to derive the parent-level predictions.
> For instance, in Table 2, in the Flat-ViT case for Living-17, the fine-level accuracy of 72.06% propagates to the coarse level, resulting in 100% consistency. However, this consistency comes at the cost of overall accuracy (e.g., FPA: 72.06, Coarse: 72.06, wAP: 72.06, TICE: 0).
>
> ---
> **(3) Variance for some key results**
>
> >(3) Given the relatively "small" sizes of the datasets by modern standards and some occasional closeness to the flat baselines in the scores (Tab 2.)  Would it be possible to include some measure of variance for some key results?
>
>  To address the reviewer’s concern about variability, we trained and Hier-ViT, and H-CAST five times on the 2-level hierarchy Living-17 dataset and the 3-level hierarchy CUB dataset and. For each run, we randomly selected 90% of the original training data and used different random seeds to ensure variability. The slight drop in performance observed is attributable to the reduced training data (90% of the full dataset). The results, presented in the table below, show that H-CAST consistently achieves strong performance with low variance, which aligns with our previously reported findings.
>
> | Living-17 | FPA           | Coarse        | Fine          | TICE         |
> |-----------|---------------|---------------|---------------|--------------|
> | Hier-ViT  | 69.71 &pm; 0.21 | 77.74 &pm;0.48 | 71.04 &pm;0.12 | 5.21 &pm;0.65 |
> | H-CAST    | 82.49 &pm;0.50 | 89.47 &pm;0.10 | 82.86 &pm;0.43 | 1.68 &pm;0.40 |
>
> | CUB      | FPA           | Order         | Family        | Species       | TICE         |
> |----------|---------------|---------------|---------------|---------------|--------------|
> | Hier-ViT | 75.48 &pm;0.13 | 98.14 &pm;0.04 | 92.78 &pm;0.24 | 77.79 &pm;0.23 | 7.13 &pm;0.52 |
> | H-CAST   | 81.21 &pm;0.39 | 98.49 &pm;0.07 | 94.48 &pm;0.39 | 83.17 &pm;0.23 | 5.28 &pm;0.40 |

---

> > ### Author Response · Authors · 2024-11-27
> >
> > Dear reviewer d3Ey,
> >
> > We hope our response has addressed your concerns. If there are any remaining questions or additional points you would like us to clarify, please let us know. Your feedback is highly valued, and we are happy to provide further explanations if needed.

---

### Author Response · Authors · 2024-11-22
**Major Updates**

Dear Reviewers and Area Chair,

We sincerely thank you for your time and effort in reviewing our manuscript. We greatly appreciate your constructive feedback and insightful comments, which have helped us strengthen our work.

As highlighted in the reviews, our work is well-motivated by the need for visual consistency in addressing inconsistent predictions for hierarchical classification. The proposed method demonstrates strong performance over baselines across benchmark datasets, supported by comprehensive experiments and analyses.

We have carefully incorporated the reviewers' comments into our revised manuscript, with all updates highlighted in blue. The major updates include:

1. A complete revision of the related work section to clearly distinguish our contributions from existing studies and to provide a more comprehensive overview of the field (Related Section, Appendix B).
2. The addition of quantitative evidence supporting our observation of the need for consistent visual grounding (Appendix A).
3. An ablation study on Tree-path KL divergence loss compared to alternative loss functions (Table 5, Appendix Table 10).
4. Visualizations of H-CAST's attention maps to illustrate how H-CAST learns visually consistent features (Appendix D.2).
5. A conclusion section, which includes a discussion of the limitations of our approach.
6. Experiments on the larger-scale dataset, iNaturalist 2021-mini (Appendix D.4.)

We believe these revisions have addressed the reviewers' comments and have further strengthened our manuscript, making it a valuable contribution to the ICLR community.

Sincerely,
The Authors

---

### Author Response · Authors · 2024-11-25
**Request for Reviewers' Feedback on Our Rebuttal and Clarifications**

Dear reviewers,

We have additionally included the results of experiments on the **larger-scale dataset, iNaturalist 2021-mini**, in Appendix D.4.

Also, we kindly ask you to review our explanation and the newly added experiments to see if they address your concerns.

Your feedback would be greatly appreciated.

Sincerely,
Authors.

---

### Comment · Reviewer_uxN6 · 2025-02-18
**The similarities between the paper and TransHP**

The similarities between the paper and TransHP are shown as below.

**The intuition.** In paper: Line 79~80; The authors argue that they NEWLY discover that the current coarse and fine-grained classifiers attend to different areas of an object. To solve this, the authors method (Fig. 1 (a)) let them have the overlap focus area. Specifically, the coarse “include” the fine. However, in TransHP, the intuitive is very similar though may not be apparent at a glance. As shown in Fig. 5 of TransHP, all the visualization shows the coarse “include” the fine. TransHP interprets this as coarse “prompts/hints” the fine. Therefore, TransHP and the paper are fundametally similar. At least, this is not NEWLY discovered by the authors.

**The realization.** In paper: Line 258 to Line 265 and the Eq. 1 is the same with TransHP: 3.3 “Multiple transformer blocks for multi-level hierarchy” and Eq. 6. Eq. 1 in the paper removes the hyperparameter of Eq. 6 in TransHP. What makes me angry is: in Line 266, authors only discuess the difference with CAST. That is certainly different, I think. But it is exactly same with TransHP. Also, in Line 267~Line 269, authors think this disign is belong to them???

**Main figure.** The Fig. 3 (left) is a horizontal adaptation version of Fig.6 combining with Fig. 4 (2) of TransHP. In this paper, there is no prompt and all the classification across different levels is performed on the same token. In TransHP, this variation (Fig. 4 (2)) is shown to be a little worse than uses prompt tokens (Fig. 4 (4)).

**Experiments.** Given the similarities above, why the authors use totally different datasets and use the excuse of no GPUs? In addition, the compared methods (FGN and HRN Line 345~Line 346) are too old: proposed in 2021 and 2022.

---

> ### Public Comment · ~Seulki_Park1 · 2025-02-22
> **Key differences between TransHP and H-CAST & Additional results of TransHP**
>
> ## **Addressing Unprofessional and Baseless Reviewer Critiques**
>
> Despite our repeated explanations and provided experiments, the reviewer continues to misrepresent our work with baseless accusations and unprofessional language, such as "*content laundering*" and "*What makes me angry is...*" A scientific review should be based on **evidence and objective critique, not emotional reactions or unfounded allegations of misconduct**.
>
> Also, accusing us of making "*the excuse of no GPUs*" is unjustified, as we clearly explained the time constraints and later provided all results.
> We are not responsible for the reviewer's frustration caused by a refusal to engage with our clarifications in good faith. **We expect reasoned, evidence-based discussion, not misinterpretations and inflammatory claims.**
>
> ------------
> ------------
> ## Before addressing the reviewer's comments, we reiterate the two key differences, as already explained in the rebuttal.
>
> ## **(1) Problem Scope**:
> **TransHP** focuses on **fine-grained prediction using coarse labels** (input: taxonomy, output: **single-level fine-grained prediction**), while **H-CAST** addresses **multi-granularity classification**, predicting the entire taxonomy (input: taxonomy, output: **whole taxonomy**). The key challenge is ensuring **consistency across levels** (e.g., avoiding mismatches like "plant" as coarse and "hummingbird" as fine).
>
> We follow prior research [1,2,3,4], using **standard benchmarks** (CUB, Aircraft, BREEDS) and evaluating **accuracy across all levels** and **consistency metrics (TICE, FPA)**. In contrast, **TransHP** focuses only on **fine-grained accuracy**, leveraging coarse labels as intermediate outputs and comparing against HiMulConE [5], which uses contrastive learning with additional coarse labels.
>
> **The differing objectives lead to distinct evaluation metrics, benchmarks, and baselines.** While both methods use hierarchical taxonomy, they belong to separate research domains, as clarified in the updated related work section.
>
> ----
>
> ## **(2) Methodology**:
> **TransHP** operates in the **semantic space**, using coarse labels as prompts to refine fine-grained classification. **H-CAST**, however, emphasizes **visual parsing consistency**, aligning visual representations with hierarchical structures across levels—from fine-grained parts to holistic scenes.
>
> H-CAST **links part-level segments to fine-grained labels** and **coarse segments to coarse labels**, ensuring unsupervised visual segmentation contributes meaningfully at each level. This shift from **semantic-space consistency (TransHP)** to **visual parsing consistency (H-CAST)** represents a fundamental methodological distinction in hierarchical classification.
>
>
> [1] Your "Flamingo" is My "Bird": Fine-Grained, or Not, 2021
> [2] Label Relation Graphs Enhanced Hierarchical Residual Network for Hierarchical Multi-Granularity Classification, 2022
> [3] Consistency-aware Feature Learning for Hierarchical Fine-grained Visual Classification, 2023
> [4] Hierarchical multi-granularity classification based on bidirectional knowledge transfer, 2024
> [5] Use All the Labels: A Hierarchical Multi-Label Contrastive Learning Framework, 2022
>
>
> ---------------------
> ## **(3) Additional results of TransHP**
> Additionally, **although the problem scope differs, we included TransHP as a ViT-based baseline** in the camera-ready version because it generates coarse labels as intermediate outputs, as requested by the reviewer.
>
> The results show that **H-CAST significantly outperforms TransHP** in **both accuracy and consistency** (FPA: **74.35% → 85.12%**, Top-1 Fine-grained Accuracy: **76.65% → 85.24%**). Meanwhile, **Hier-ViT**, a TransHP variant (Fig. 4 (2)), performs slightly worse than TransHP, supporting their claim.
>
> | Living-17 | FPA   | Coarse | Fine  | wAP   | TICE  |
> |-----------|-------|--------|-------|-------|-------|
> | HRN       | 79.18 | 87.53  | 81.47 | 83.49 | 6.29  |
> | Hier-ViT  | 74.06 | 80.94  | 74.88 | 76.90 | 10.50 |
> | TransHP   | 74.35 | 83.00  | 76.65 | 78.76 | 8.35  |
> | **H-CAST**    | **85.12** | **90.82**  | **85.24** | **87.10** | **3.19**  |
>
> Furthermore, on **iNaturalist-2018, the exact dataset used in TransHP**, H-CAST achieves **strong top-1 accuracy**, confirming its effectiveness. (Here, H-CAST was trained as a small model for 100 epochs.)
> |           | iNat-2018 |
> |-----------|:---------:|
> | Guided    |   63.11   |
> | HiMulConE |   63.46   |
> | TransHP   |   64.21   |
> | **H-CAST**    |   **67.13**   |
>
> **Thus, H-CAST is NOT a variant of TransHP, and the consistent visual grounding and TK loss we introduce are highly effective.**
>
> For optimal setup, we used the official codebase, training for 300 epochs (H-CAST: 100 epochs). Prompt block placement for coarse-level supervision followed Table 1 in the TransHP paper:
> - 2-level datasets: Selected the better-performing configuration between [6, 11] and [8, 11].
> - 3-level datasets: Used [6, 8, 11] blocks.

---

> ### Public Comment · ~Seulki_Park1 · 2025-02-22
> **Comments for the reviewer's concerns**
>
> ## **(1) The intuition**.
>
> The claim that our intuition is the same as TransHP is incorrect. **TransHP encourages semantic consistency through hierarchical prompting but does not enforce visual consistency.** Additionally, Fig. 5 in TransHP only visualizes correctly predicted cases, but this is not unique to TransHP—prior works, including [6] (See Fig. 5), also show correct cases.
>
> In contrast, we **specifically investigate when inconsistencies occur and how to resolve them.** Our analysis reveals that **coarse and fine classifiers often attend to entirely different regions for the same image**, leading to misaligned predictions. To address this, we **explicitly encourages visual consistency**, ensuring that classifiers remain visually aligned across hierarchy levels. Rather than just enforcing semantic consistency, we **leverage visual segments to correct inconsistencies**, making our approach fundamentally different.
>
> **Thus, our intuition is distinct from TransHP, as we focus on visual grounding rather than semantic refinement.**
>
>
> [6] Where to Focus: Investigating Hierarchical Attention Relationship for Fine-Grained Visual Classification, ECCV, 2022.
>
>
> -------
>
> ## **(2) The realization**.
>
> (1) Line 258-265 and Equation (1): Using cross-entropy loss for hierarchical supervision is a widely adopted practice [Equation (3) in [6], Equation (1) in [7]]. However, the uniqueness of each approach lies in its architectural design, block structure, feature utilization, and additional loss formulation. Claiming that two methods are identical solely based on shared equations oversimplifies the distinctions and fails to acknowledge these critical differences.
>
> (2) Regarding TransHP, we have already discussed its relevance in the Related Work section, where we believe it is most appropriate. In the camera-ready version, we have further expanded the discussion on TransHP in the Experimental section to provide additional clarity.
>
> (3) Line 266 (Methodology section): The discussion focuses on the difference between our method (H-CAST) and CAST because H-CAST adopts CAST's architecture to leverage visual segments. Since our methodology is built upon this design choice, CAST is the most relevant comparison in this section.
>
> (4) Lines 267–269: Your concern is unclear. This section explains our design choice—Fine-to-Coarse supervision, which contrasts with prior methods ([6], [7], and even TransHP) that follow the Coarse-to-Fine direction. The intent is to highlight this methodological difference, not to claim ownership of a general concept.
>
>
> [6] Where to Focus: Investigating Hierarchical Attention Relationship for Fine-Grained Visual Classification, ECCV, 2022.
> [7] B-CNN: Branch Convolutional Neural Network for Hierarchical Classification, 2017.
>
> ----------
>
> ## **(3) Main figure**
> As previously explained during the rebuttal, **Figure 4 (2) in TransHP aligns more closely with Hier-ViT, another baseline we consider, rather than H-CAST**. Unlike H-CAST, **it lacks visual segments and TK loss—key components of our method**.
>
> The difference is evident in performance: **H-CAST improves accuracy by +2.92pp on iNat-18, nearly four times TransHP’s +0.75pp gain over the previous SOTA**. This demonstrates the effectiveness of visual grounding with segments and TK loss.
>
> Thus, equating H-CAST with TransHP’s Figure 4 (2) is inaccurate, as our methodological differences lead to significantly stronger performance.
>
> |           | iNat-2018 |
> |-----------|:---------:|
> | Guided    |   63.11   |
> | HiMulConE |   63.46   |
> | TransHP   |   64.21   |
> | H-CAST    |   67.13   |

---

> ### Public Comment · ~Seulki_Park1 · 2025-02-22
> **Comments for the reviewer's concerns**
>
> ## **(4) Experiments**
>
> → (1) **We did not avoid specific datasets due to GPU limitations.** During the review period, we mentioned that large-scale dataset experiments would **take longer due to GPU constraints** and requested patience so that we could first proceed with the discussion. **A few days later, we updated the results**. Below is the exact statement we provided at that time:
>
> *"We are currently conducting additional experiments on the large-scale iNaturalist dataset to further validate our method on a larger dataset. We will update the results as they become available. However, we kindly ask for your understanding, as our training environment relies on a single GPU (Nvidia A40), which causes the experiments to take longer to complete."*
>
>
>
> (2) As discussed earlier, our task is fundamentally **different from single-level fine-grained classification**. We focus on **multi-granularity classification**, which requires different benchmark datasets. Thus, **we followed prior work in this research line and adopted the standard datasets used in this area**.
>
> (3) Additionally, **FGN and HRN remain strong baselines in this field**. They are not simply "*old methods*"—both have demonstrated competitive performance and have publicly available codebases. As shown in our results, **HRN even outperforms TransHP**. Additionally, more recent works in multi-granularity classification ([3], [4]) have not released their code, making FGN and HRN the most practical and reproducible baselines for comparison.
>
>
> | Living-17 | FPA   | Coarse | Fine  | wAP   | TICE  |
> |-----------|-------|--------|-------|-------|-------|
> | HRN       | 79.18 | 87.53  | 81.47 | 83.49 | 6.29  |
> | TransHP   | 74.35 | 83.00  | 76.65 | 78.76 | 8.35  |
> | H-CAST    | 85.12 | 90.82  | 85.24 | 87.10 | 3.19  |
>
>
> [3] Consistency-aware Feature Learning for Hierarchical Fine-grained Visual Classification, 2023
> [4] Hierarchical multi-granularity classification based on bidirectional knowledge transfer, 2024

---

### Meta-Review · Area_Chair_ujzR · 2024-12-16

**Metareview:**

The paper proposes an architecture that extends CAST for hierarchical classification tasks. The proposal fuses superpixels of high similarity using a graph-pooling operation within the ViT tokens. The hierarchical classification is achieved through a set of classification heads per level in the hierarchy.

Strengths:
- Improved results over existing works
- Introduction of new metrics for hierarchical classification
- Reported significant improvements in the experiments
- Ablations confirm the contribution of the added losses

Weaknesses:
- The primary contributions of this work are merely the hierarchical supervision loss and tree KL loss, neither of which can ensure visual consistency
- It is unclear why the model would “ensure that each hierarchical classifier focuses on the same corresponding regions”
- The experiments are somewhat questionable; the reviewers asked to compare against newer methods, but the authors mentioned that the used methods are relevant
- The experiments do not fully compare against the most recent methods
- The paper should be updated to include the latest comparisons and discuss the overlap with existing methods such as CAST and TransHP

The paper received mixed reviews with critical comments due to the limited technical contributions, mainly because it uses existing methods such as CAST, and its similarity to existing approaches like TransHP. I agree with the authors that there are nuanced differences between the approaches and that demonstrating the effectiveness of the proposal in these settings is a contribution in itself. Moreover, as some reviewers mentioned, the method shows improvements over existing methods. While it would be interesting to see experiments on larger datasets, the authors' rationale for selecting the datasets used is sound and follows the literature on hierarchical classification. I also do not agree with requesting more experiments merely for the sake of experimentation. Utilizing existing methods in a new way to exploit instance-level classification and demonstrate its advantages is a contribution in itself. Thus, I recommend the acceptance of the paper.

**Additional Comments On Reviewer Discussion:**

Reviewer d3Ey identified the strengths of the paper as solid experimental results that support its claims. However, the setup was criticized due to the lack of labels at coarser levels. The authors addressed the reviewer's comments and included the requested experiments, but the reviewer did not reply further.

Reviewer khL4 commented that the visual consistency is sound and that the proposal performs well on the evaluated datasets. The paper also introduces metrics for hierarchical classification. However, the reviewer raised concerns about the lack of theory to support the idea of attending to visual cues at different levels. They also noted that the paper relies heavily on CAST and that the main contribution is incremental. The reviewer was concerned about the comparisons, as they do not include more recent methods. Although the authors replied to the reviewer's comments, the reviewer remained unconvinced and stated that the work does not meet ICLR standards, expressing a need for comparisons against more recent methods. The reviewer questioned the differentiation between part-to-whole and taxonomy hierarchies, despite the authors' explanations.

Reviewer YARY had a positive view of the paper, finding it well-written, organized, and sound, with experiments that demonstrate the contributions. This reviewer raised questions about the use of coarse-to-fine labels and whether the approach could be resolved by better predictions at lower levels. The authors addressed these questions to the reviewer's satisfaction.

Reviewer ZNGF noted considerable improvements over the baselines but raised issues with the technical contributions, which build on CAST. The reviewer suggested that experiments could include pixel-level classification as well. Concerns were also raised about the omission of leading methods in the evaluation. The authors responded to the reviewer's concerns, but the reviewer felt that the novelty and evaluation were not fully convincing.

Reviewer uxN6 was highly critical, stating that the paper heavily builds on TransHP and is very similar to it. The reviewer also mentioned that the proposal uses established techniques and lacks sufficient explanation of technical contributions. Despite the authors’ responses, the reviewer maintained their concerns.

After the rebuttal, the authors reached out to comment on the adversarial stances of reviewers uxN6 and khL4. Reviewer uxN6 had exhibited an adversarial stance from the beginning, including an extremely aggressive and unprofessional initial review, which was later updated. Reviewer khL4 adopted a similar stance by the end of the exchanges and did not provide additional justification for their claim that the paper is subpar.

During the post-rebuttal discussion, reviewers khL4 and uxN6 reiterated their stance to reject the paper. However, the most positive reviewer, YARY, maintained that while the architecture and methods are not entirely novel, the authors identified and exploited an intrinsic potential for hierarchical classification. YARY stated that the improvements over the baselines are justified.

Given the extremely negative reviewers' adversarial perspectives, I am more inclined to give greater weight to the positive contributions highlighted by YARY. Thus, while I recommend the paper for acceptance, I am not fully convinced, given the raised issues and limited contribution.

---

### Decision · Program_Chairs · 2025-01-22

Accept (Poster)